# General variation in the *Fusarium* wilt rhizosphere microbiome

Lv Su [1,2,6], Haichao Feng[3,6], Huatai Li[4], Fu Yang[4], Xiaoqian Yan[2,3], Xudong Wang[2,5], Pengcheng Li[2], Kesu Wang[1], Xia Shu[4], Yunpeng Liu [4], Qirong Shen [1], Yongliang Yan [2] ✉ & Ruifu Zhang [1] ✉

The dominant bacteria enriched in the *Fusarium* wilt plants' rhizosphere are of increasing interest, as they adapt well to the diseased rhizosphere. However, general information about these bacteria is still lacking. Here, we perform a meta-analysis of *Fusarium* wilt plants rhizosphere and comprehensive studies to obtain information about the robust variation in the rhizosphere microbiome of *Fusarium* wilt plants. We demonstrate that *Fusarium* infection reproducibly changes the rhizosphere bacterial community composition. The rhizosphere microbiomes of *Fusarium* wilt plants are characterized by the enrichment of *Flavobacterium*, gene cassettes involved in antioxidant functions related to sulfur metabolism and the root secreted tocopherol acetate. We further isolate antagonistic *Flavobacterium anhuiense* from the diseased tomato rhizosphere, and reveal that the growth of *F. anhuiense* and the expression of genes related to carbohydrate metabolism in this strain are significantly stimulated by tocopherol acetate. Furthermore, the inhibitory effect of *F. anhuiense* against *F. oxysporum* and *F. anhuiense* population enhancement by tocopherol acetate are confirmed *in planta*. The robust variation in the rhizosphere microbiome elucidates key principles governing the general assembly mechanism of the microbiome in the *Fusarium* wilt plants' rhizosphere.

The plant microbiome is referred to as the second genome of a plant and plays important roles in host health[1]. Imbalance of the plant microbiome, known as dysbiosis, significantly influences plant health[2]. The disruption of potential antagonistic bacteria, such as Firmicutes and Actinobacteria, in the tomato rhizosphere by vancomycin may lead to microbiome dysbiosis, potentially enriching pathogens such as *Ralstonia solanacearum* and *Fusarium oxysporum*[3]. Similarly, invasion by the tomato soil-borne pathogen *R. solanacearum* is found in many rhizosphere ecological niches, significantly decreasing the relative abundances of antagonistic bacteria

and the bacterial diversity that may contribute to other fungal pathogen infections[4]. Plants and microbes have coevolved for hundreds of millions of years and formed a very sophisticated and beneficial system of communication. Certain beneficial bacteria can be enriched in the rhizosphere of plants infected with *Fusarium* pathogens, potentially contributing to disease suppression or plant health improvement[5,6]. *Bacillus* and *Sphingomonas* are significantly enriched in the *Fusarium* wilt rhizosphere and protect cucumber from pathogen infection by increasing the level of reactive oxygen species in the roots[7]. Similarly, the *Fusarium*-infected cabbage

[1]College of Resources and Environmental Sciences, Nanjing Agricultural University, Nanjing, Jiangsu, The People's Republic of China. [2]Biotechnology Research Institute/National Key Laboratory of Agricultural Microbiology, Chinese Academy of Agricultural Sciences, Beijing, China. [3]College of Agriculture, Henan University, Kaifeng, Henan, China. [4]State Key Laboratory of Efficient Utilization of Arid and Semi-arid Arable Land in Northern China, The Institute of Agricultural Resources and Regional Planning, Chinese Academy of Agricultural Sciences, Beijing, China. [5]College of Life Sciences, Yangzhou University, Yangzhou, Jiangsu, China. [6]These authors contributed equally: Lv Su, Haichao Feng. ✉e-mail: yanyongliang@caas.cn; rfzhang@njau.edu.cn

recruited beneficial *Pseudomonas*, which significantly suppressed pathogen growth and enhanced plant disease resistance[8]. These findings indicate that certain general changes may exist in the rhizosphere microbiome of diseased plants. Understanding such general changes will provide deep insights into the recruitment mechanism of rhizosphere bacterial communities.

Elucidation of the microbial recruitment mechanism can help clarify the causes of changes in diseased rhizosphere microbial composition, which is an important topic in microbial ecology. Root exudates[9], phytohormones[10], quorum sensing[11], induced systemic resistance[12], and nutrient availability[13] significantly influence the recruitment of rhizosphere bacterial communities. Among these, root exudates serve as a critical bridge linking plants and rhizosphere microbes. Most of the pathways discussed above involve the influence of root exudates on microbe recruitment. During this process, microbes are initially attracted by specific components of root exudates and colonize the root surface through chemotaxis. Subsequently, they utilize root exudates for rapid proliferation and establish stable colonization on the rhizoplane by forming biofilms[14]. Moreover, pathogen invasion could significantly alter the composition of root exudates. For example, the metabolites such as tocopherol acetate, citrulline, galactitol, octadecylglycerol, and behenic acid were greatly enriched in the rhizosphere of cucumber infected by *Fusarium*[15]. *Fusarium* infection also triggered succinic acid and oxalic acid production in the watermelon root exudates[16]. These enriched metabolites may contribute to the rhizosphere recruitment of beneficial microbes, which protect plants from pathogen invasion. For instance, *Arabidopsis thaliana* infected by *P. syringae pv* tomato induces the secretion of malic acid that stimulates biofilm formation of *Bacillus subtilis*[17]. Furthermore, pathogen invasion can alter the functional profile of the rhizosphere microbiome, reflecting changes in the genes related to nutrient metabolism. The relative abundances of genes associated with amino acid transport and metabolism decreased in the rhizosphere of *Fusarium*-wilt *P. notoginseng*, suggesting a depletion of microbes capable of utilizing these metabolites[18]. Overall, the combined application of metabolomics and metagenomics provides a systematic approach to unravel the recruitment mechanisms of rhizosphere microbes, offering deep insights into plant-microbe interactions under pathogen stress.

Root rot diseases caused by *Fusarium* severely threaten crop yields worldwide[19]. Owing to their environmentally friendly characteristics, beneficial microbes have become an alternative method for the control of *Fusarium* wilt diseases. However, colonization by beneficial microbes is easily affected by the soil environment and cannot play a stable role in the control of *Fusarium* wilt diseases[20]. Identification of the microbes that are common in the *Fusarium* wilt rhizosphere from different soils may reveal potential beneficial strains.

Here, we performed a meta-analysis of *Fusarium* wilt rhizosphere studies and comprehensive experiments to investigate the cross-kingdom general effects of *Fusarium* infections on the bacterial community in the rhizosphere. Despite major differences in experimental design between independent studies, we identified reproducible signatures in which *Flavobacterium*, antioxidant functions related to sulfur metabolism and the root secreted tocopherol acetate were significantly enriched in diseased rhizospheres. We experimentally validated that *Fusarium* infections significantly affected rhizosphere bacterial communities and enriched *Flavobacterium* and tocopherol acetate. In addition, the antagonistic microbe *Flavobacterium anhuiense* K5 was isolated from the diseased tomato rhizosphere, and its growth and carbohydrate metabolism were significantly stimulated by tocopherol acetate.

## Results
### Study selection and characteristics
A total of 37 amplicon and four metagenomic studies were retrieved via our search methodology. Studies investigating changes in the

fungal community (seven amplicon studies) were excluded from the analysis, as most of them lack sufficient metadata information. Among the remaining bacterial amplicon studies, six studies do not pass the sequence quality control (QC) process, nine studies lack sufficient sequencing metadata (e.g., lacking essential information such as treatment and control details, which made it impossible to distinguish between healthy and diseased plants), and six studies for other diseases were excluded (two studies for *Ralstonia solanacearum* bacterial wilt, two studies for nematode diseases and two studies for fungal diseases caused by *Rhizoctonia* and *Cylindrocarpon*). The amplicon of *Fusarium* wilt from the remaining 9 studies was used to determine the changes in the rhizosphere bacterial community caused by *Fusarium* infection (Supplementary Data 1). These amplicon studies of *Fusarium* infections include 198 rhizosphere samples (93 diseased samples and 105 healthy samples) from 18 study areas. These studies varied considerably in terms of the host types (Supplementary Table 1). These samples were obtained from 7 plants (pea, panax, pepper, tomato, avocado, watermelon and cucumber). In total, 198 samples were included in our meta-analysis (Supplementary Table 2). Among the 4 metagenomic studies, 2 studies of root endophytes were removed. These metagenomic studies of *Fusarium* infections include 22 rhizosphere samples (11 diseased samples and 11 healthy samples) from 3 study areas. The metagenomic samples were obtained from two plants (panax and zanthoxylum).

### *Fusarium* infection reproducibly changes rhizosphere bacterial community composition
We calculated common metrics for alpha diversity. No significant or consistent changes in Shannon ($P = 0.06$) and Simpson ($P = 0.84$) diversity were observed between diseased and healthy rhizospheres via the linear mixed effects model. However, there was a modest increase in observed richness, ACE, Chao, and Faith phylogenetic diversity indices in the bacterial community of the diseased rhizosphere ($P < 0.001$) (Supplementary Fig. 1).

We employed visualization strategies to reveal the rhizosphere bacterial community composition via principal coordinate analysis, and tested the effects of *Fusarium* infection on community composition at each operational analysis unit (OAU), which refines the analysis by dividing these samples into subgroups. This allows comparisons between diseased and healthy rhizosphere microbiomes under consistent conditions (e.g., same sampling location and time within a study). In the present study, the 198 samples were divided into 25 OAUs. Significant differences in the bacterial composition between the diseased and healthy plants' rhizosphere bacterial communities were observed in 11 OAUs of 25 OAUs (Fig. 1, Supplementary Figs. 2 and 3, Supplementary Data 2 and 3). In addition, although there were no significant differences in the bacterial compositions, obvious differences were found in three OAUs (OAU21, OAU22, and OAU23). Considering all these studies together, a significant effect of *Fusarium* infection on community composition was revealed with a linear mixed effects model ($P < 0.001$, $F = 17.7$) (Supplementary Fig. 3). Given the clear evidence for disease signals from multivariate analysis, we further determined which specific features of the microbiota are responsive to *Fusarium* infection (Supplementary Fig. 4).

A random forest classifier was used to define the biomarkers of the bacterial response to *Fusarium* infection (Fig. 2A). The top 20 most-predictive genera were selected based on the MeanDecreaseGini or MeanDecreaseAccuracy indices, and these genera of *Stenotrophomonas*, *Sphingobacterium*, *Variovorax*, *Stella*, *Flavobacterium*, *Aeromonas*, *Luteimonas*, *Sporocytophaga*, *Novosphingobium* were significant predictors in both indices.

To obtain stable and reliable biomarkers, we selected the top predictive genera that exhibited significant statistical relevance in the MeanDecreaseGini or MeanDecreaseAccuracy index (31 genera in total) and performed another round of prediction using the automated

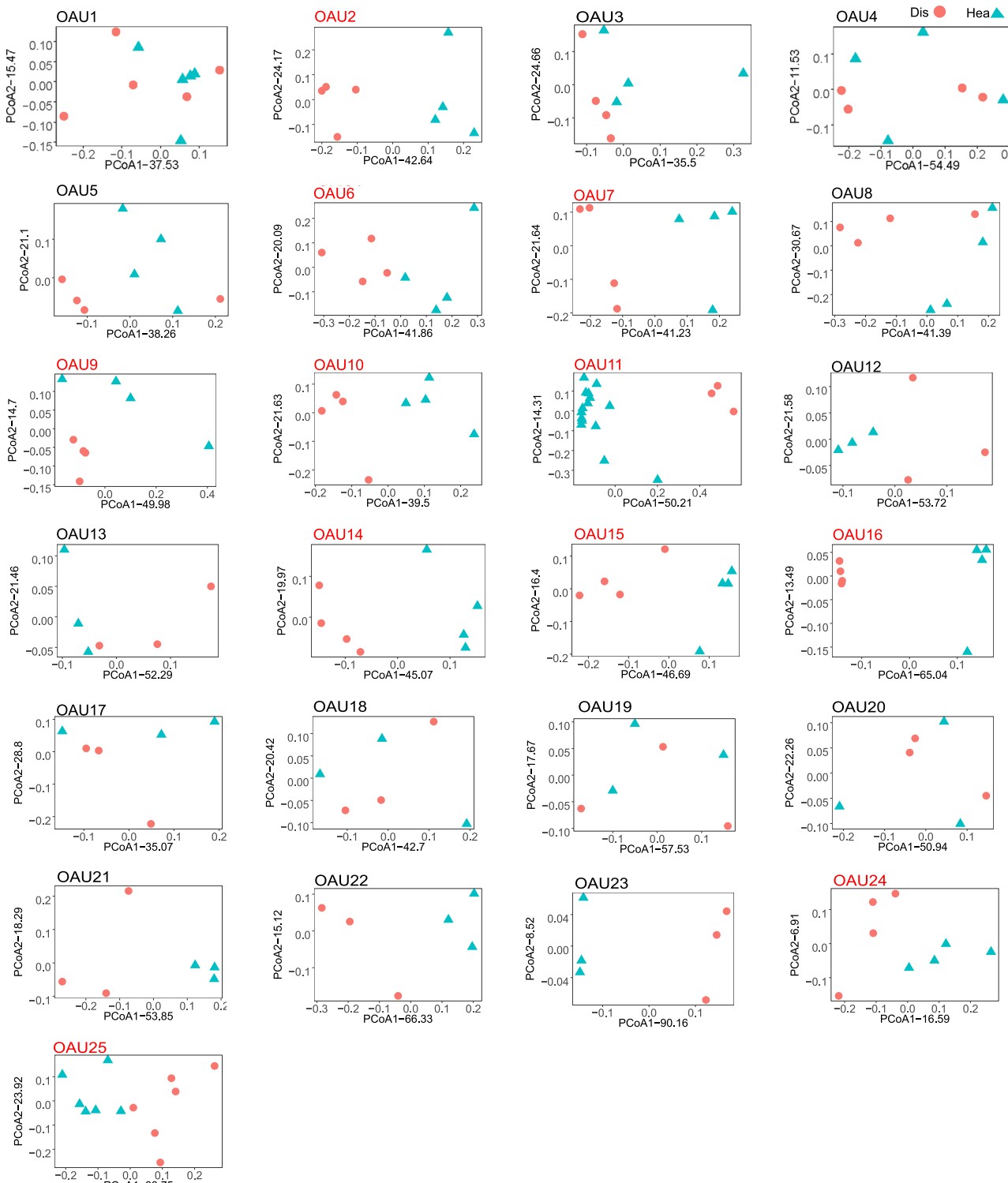

**Fig. 1 | Beta-diversity analysis of the rhizosphere bacterial community based on weighted UniFrac distance at ASV level in OAU (operational analysis units).** The OAU represents the subgroups whose disease and healthy sample groups could be compared under the same conditions. The OAUs that contained significantly distinct clusters between diseased and healthy communities are indicated in red. Statistical significances between diseased and healthy communities in OAUs were determined based on the anosim function in the vegan package. All analyses were performed on $n = 93$ and 105 for diseased and healthy independent rhizosphere samples. Source data are provided as a Source Data file.

hyperparameter-tuning mlr machine learning framework with the random forest algorithm. The accuracy of this prediction model was increased by 10% compared to the initial predictive criteria, reaching 81% (Supplementary Fig. 5). The results demonstrated that *Stenotrophomonas*, *Sphingobacterium*, *Variovorax*, *Stella*, *Flavobacterium*, *Luteimonas*, and *Sporocytophaga* were among the top 10 most predictive genera (Supplementary Table 3). This finding was consistent with the genera that showed statistical significance in both the MeanDecreaseGini and MeanDecreaseAccuracy indices, confirming the robustness of our selection methodology. SHAP (SHapley Additive

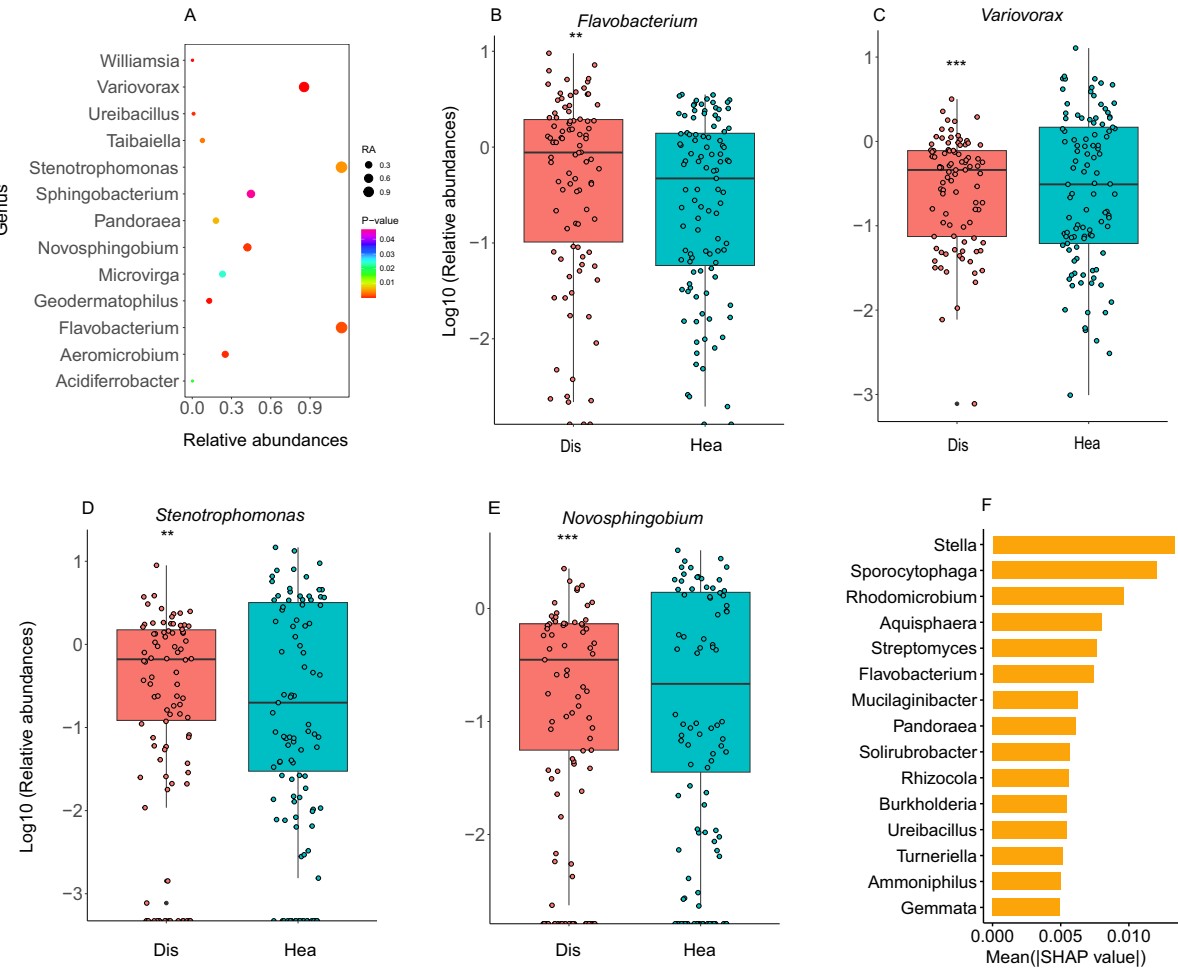

**Fig. 2 | A reproducible signature of the *Fusarium* infection rhizosphere microbiome. A** Relative abundance of the significant predictors in the Mean-DecreaseGini or MeanDecreaseAccuracy indices from the random forest analysis. The significantly changed genera are colored according to the significance level. The *x* axis label represents the mean across all studies. **B**–**E** Significantly altered biomarkers between the diseased and healthy rhizosphere microbiomes. These biomarkers were identified based on predictors that showed significance in both the MeanDecreaseGini and MeanDecreaseAccuracy indices. Data are presented as box plots, where the center line indicates the median, the box limits indicate the upper and lower quartiles (25th and 75th percentiles), and the whiskers extend to the minimum and maximum data points. Statistical significance was determined using a linear mixed effects model with the formula: relative abundance -health_-status + (1 | OAU), followed by ANOVA: *Variovorax* ($P = 4.00 \times 10^{-5}$), *Novo-sphingobium* ($P = 7.00 \times 10^{-4}$), *Stenotrophomonas* ($P = 4.90 \times 10^{-3}$), *Flavobacterium* ($P = 1.50 \times 10^{-3}$). **F** SHapley Additive exPlanations (SHAP) value analysis of rhizosphere microbiome composition comparing diseased and healthy samples. The plot displays the top 15 most discriminative bacterial genera ranked by their mean absolute SHAP values. All analyses were performed on $n = 93$ and 105 for diseased and healthy independent rhizosphere samples. *** $P < 0.001$, ** $P < 0.01$, * $P < 0.05$. Source data are provided as a Source Data file.

exPlanations) analysis was employed to evaluate feature importance. These results also identified *Stella*, *Flavobacterium*, *Sphingobacterium*, and *Sporocytophaga* as the top predictive genera (Fig. 2F). We further used a linear mixed effects model to determine the significantly changed genera among these biomarkers; results showed that *Variovorax*, *Flavobacterium, Stenotrophomonas*, and *Novosphingobium* were significantly enriched in the diseased rhizosphere. Moreover, *Flavobacterium* and *Stenotrophomonas* were the potential beneficial bacteria that could inhibit the growth of *Fusarium* (Supplementary Table 4). Overall, *Flavobacterium* is the only genus consistently identified as a biomarker by multiple analytical methods (Fig. 2B and Fig. 2F), and it additionally exhibits potential biocontrol capabilities.

To verify the meta-analysis results, we collected *Fusarium*-diseased and healthy rhizosphere soil samples from tomato and *Panax notoginseng* plants (Fig. 3). Compared to the *Fusarium*-diseased rhizosphere, the Shannon and Faith phylogenetic diversity indices were significantly decreased and increased in the healthy tomato (Supplementary Fig. 6A) and *Panax notoginseng* (Supplementary Fig. 6B) rhizosphere soil samples, respectively. Moreover, *Fusarium* infection

significantly changed the bacterial community in the tomato rhizosphere (anosim test $P < 0.05$, betadisper test $P > 0.05$). Obvious differences were found in the diseased and healthy rhizosphere bacterial communities of *P. notoginseng* plants (anosim test $P < 0.05$, betadisper test $P < 0.05$).

The different changes in bacterial community composition were determined at the ASV level. Like the metadata results, the dominant *Variovorax* (ASV5), *Flavobacterium* (ASV36), *Stenotrophomonas* (ASV434), and *Novosphingobium* (ASV133) were significantly enriched in the *P. notoginseng*-diseased rhizosphere. The dominant *Flavo-bacterium* (ASV60) and rare *Novosphingobium* (ASV847) were also significantly enriched in the diseased tomato rhizosphere. Because *Flavobacterium* is the only dominant genus significantly enriched in both the metadata and validation experiment, two *Flavobacterium* isolates, K1 and K5, were isolated from the diseased tomato rhizosphere soil (from a total of 36 isolates). The 16S rRNA of strains K1 and K5 were matched to the sequences of ASV60 (relative abundance: 0.22%) and ASV15936 (relative abundance: 0.006%) at 97% similarity. The dominant bacterial strain K5 (*Flavobacterium anhuiense*) was used

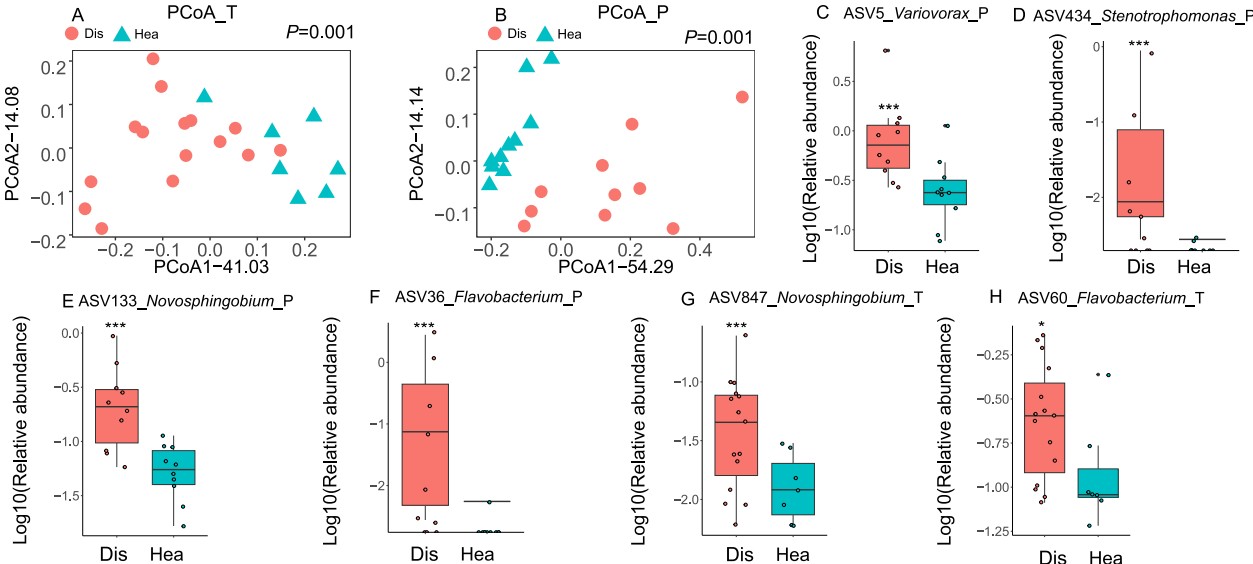

**Fig. 3 | Differences in the rhizosphere microbiome of tomato and *Panax notoginseng* plants between *Fusarium*-diseased and healthy conditions.** Beta-diversity analysis of the rhizosphere bacterial communities from tomato (**A**) and *Panax notoginseng* (**B**) plants based on weighted UniFrac distances. **C–H** Significantly enriched ASVs in the diseased rhizosphere microbiome of *Panax notoginseng* (**C–F**) and tomato (**G, H**). Data are presented as box plots (center line, median; box limits, upper and lower quartiles (25th/75th percentiles); whiskers, min/max). Each data point represents an independent biological sample.

Differential abundance was tested using the DESeq2 package (Benjamini–Hochberg FDR correction). Exact FDR-adjusted *P*-values for the displayed ASVs are: ASV5_P = 3.61 × 10⁻⁵, ASV434_P = 4.69 × 10⁻⁸, ASV133_P = 2.39 × 10⁻⁸, ASV36_P = 4.08 × 10⁻¹⁴, ASV847_T = 9.63 × 10⁻³, ASV60_T = 4.15 × 10⁻². \*\*\**P* < 0.001, \*\**P* < 0.01, \**P* < 0.05. Analysis in **A**, **G**, and **H** was performed on *n* = 15 diseased and *n* = 7 healthy independent rhizosphere samples. Analysis in **B–F** was performed on *n* = 10 independent rhizosphere samples. Source data are provided as a Source Data file.

to test its effects on the growth of *F. oxysporum*, revealing a significant inhibition of the pathogen growth (Supplementary Fig. 7).

## *Fusarium* infection reproducibly changes rhizosphere microbiome functions

We next determined the general changes in the microbiome functions of the diseased rhizosphere (Table 1 and Table 2). A Kyoto Encyclopedia of Genes and Genomes (KEGG) ortholog (a set of orthologous genes that are functionally conserved across different organisms) encoding thioredoxin-dependent peroxiredoxin (K03564), associated with antioxidant functions (Supplementary Table 5), was markedly enriched in the diseased rhizosphere. Similarly, KEGG orthologs involved in sulfur metabolism, including tRNA-uridine 2-sulfurtransferase (K00566), sulfur-carrier protein (K03636), and sulfur-carrier protein adenyltransferase (K21147), also showed increased abundance in the diseased rhizosphere (Supplementary Table 6). In contrast, KEGG orthologs linked to root exudate metabolism, such as malate dehydrogenase (K00029), fumarate hydratase (K01679), and dihydroxy-acid dehydratase (K01687), were more abundant in the healthy rhizosphere (Supplementary Table 7). Notably, KEGG orthologs associated with chitin degradation, including acetylhexosaminidase (K01207), were highly enriched in the diseased rhizosphere. Furthermore, KEGG orthologs related to stress responses, such as integrase (K04763), phosphate regulon sensor histidine kinase (K07636), and NAD(P)H-hydrate dehydratase (K17759), were also elevated in the diseased rhizosphere. Moreover, several of these KEGG orthologs (e.g., K01207, acetylhexosaminidase; K04763, integrase; and K07636, phosphate regulon sensor histidine kinase) were identified in the *F. anhuiense* K5 genome (Supplementary Table 8).

## Root secreted tocopherol acetate enriches strain *F. anhuiense* K5

To identify the root exudate components that drive the enrichment of *F. anhuiense* K5, the root exudate components from the *Fusarium*-infected plants in 67 published studies were collected (Supplementary

Data 4). We found that 32 root exudate components, such as chlorogenic acid, arginine, nicotinic acid, fumaric acid, glutamic acid, and tocopherol acetate were significantly enriched in the *Fusarium*-infected plants (Supplementary Data 5). In addition, an experiment was performed to collect tomato root exudates from *Fusarium*-infected and healthy plants. The metabolome results showed that *Fusarium* invasion significantly affected the root exudate composition (Fig. 4). Analysis of the top 50 components revealed distinct biomarker profiles between plant health statuses. Diseased plants showed significant enrichment of four root exudate components: tocopherol acetate (Supplementary Fig. 8), o-toluic acid, phenylacetamide, and 7-methylguanine, while healthy plants exhibited elevated levels of 13 different root exudates (Supplementary Table 9)

A total of 35 enriched root exudate compounds from diseased plants were investigated to determine their effects on the growth of strain K5. We found that 27 compounds significantly stimulated the growth of strain K5 at 19 h (the stationary phase). Specifically, the highest stimulation level was found for tocopherol acetate (-1.24-fold greater than controls). Similar results were also found at 12 h (logarithmic phase) and 30 h (decay phase), where the cell density (OD₆₀₀) for tocopherol acetate was 1.19 and 1.25-fold greater than the controls, respectively. Moreover, at 12 h and 30 h, the numbers of root exudates components that significantly stimulated the growth of strain K5 were 16 and 14, respectively (Supplementary Fig. 9 and Supplementary Fig. 10). We further determined the effects of different concentrations of tocopherol acetate on strain K5 at 12 h (Fig. 5A). The results showed that 0.1 mM tocopherol acetate significantly stimulated the growth of strain K5, and the maximum stimulation concentration was 1 mM.

We also determined the effects of tocopherol acetate on the soil bacterial community. The results revealed that 1 mmol/kg tocopherol acetate significantly changed the bacterial community structure (Fig. 5B). The genus *Flavobacterium* was significantly enriched in the soil supplemented with tocopherol acetate (Supplementary Table 10). In addition, the sequence of ASV1, whose sequence was matched to the

**Table 1 | Significantly co-enriched genes in the diseased rhizosphere from the metagenome analysis of different OAUs**

| KO | Function | Dis | Hea | P | Fold (D/H) | OAU |
|---|---|---|---|---|---|---|
| K00012 | UDPglucose 6-dehydrogenase | 367.8 ± 34.41 | 359.12 ± 8.21 | 0.0831 | 1.02 | OAU1 |
| | | 377.33 ± 8.35 | 312.58 ± 54.16 | | 1.21 | OAU2 |
| K01207 | beta-N-acetylhexosaminidase | 296.23 ± 2.83 | 263.73 ± 22.09 | 0.0124 | 1.12 | OAU1 |
| | | 265.89 ± 35.9 | 232.13 ± 28.64 | | 1.15 | OAU2 |
| K01754 | threonine dehydratase | 596.13 ± 36.23 | 584.2 ± 13.07 | 0.0989 | 1.02 | OAU1 |
| | | 558.57 ± 36.55 | 488.66 ± 56.07 | | 1.14 | OAU2 |
| K00566 | tRNA-uridine 2-sulfurtransferase | 241.74 ± 22.59 | 236.16 ± 9.36 | 0.2903 | 1.02 | OAU1 |
| | | 189.26 ± 32.32 | 164.52 ± 30.58 | | 1.15 | OAU3 |
| K03564 | thioredoxin-dependent peroxiredoxin | 360.69 ± 23.12 | 329.19 ± 8.87 | 0.0249 | 1.1 | OAU1 |
| | | 401.72 ± 97.34 | 267.18 ± 69.57 | | 1.50 | OAU2 |
| K03636 | sulfur-carrier protein | 252.93 ± 43.47 | 198.77 ± 17.05 | 0.0130 | 1.27 | OAU1 |
| | | 295.88 ± 77.48 | 207.71 ± 32.01 | | 1.42 | OAU2 |
| K05710 | trans-cinnamate dioxygenase ferredoxin component | 192.98 ± 20.17 | 177.83 ± 17.46 | 0.1135 | 1.09 | OAU1 |
| | | 265.44 ± 104.17 | 156.54 ± 69.77 | | 1.70 | OAU2 |
| K21147 | sulfur-carrier protein adenylyl transferase | 246.83 ± 24.48 | 228.97 ± 25.31 | 0.1605 | 1.08 | OAU1 |
| | | 214.37 ± 47.29 | 182.69 ± 34.23 | | 1.17 | OAU3 |
| K04763 | integrase/recombinase | 374.98 ± 18.76 | 366.53 ± 19.32 | 0.1228 | 1.02 | OAU1 |
| | | 463.79 ± 117.71 | 326.01 ± 84.98 | | 1.42 | OAU2 |
| K07636 | phosphate regulon sensor histidine kinase | 348.11 ± 20.71 | 308.85 ± 25.66 | 0.0544 | 1.13 | OAU1 |
| | | 287.11 ± 46.43 | 261.6 ± 39.91 | | 1.10 | OAU3 |
| K17758 | ADP-dependent NAD(P)H-hydrate dehydratase | 222.13 ± 17.84 | 198.12 ± 15.45 | 0.0199 | 1.12 | OAU1 |
| | | 204.51 ± 31.09 | 129.12 ± 56.26 | | 1.58 | OAU2 |
| K17759 | NAD(P)H-hydrate epimerase | 215.21 ± 19.58 | 189.03 ± 13.67 | 0.0186 | 1.14 | OAU1 |
| | | 199.51 ± 30.75 | 120.15 ± 60.93 | | 1.66 | OAU3 |

Operational analysis unit (OAU) represents a subgroup from a study with multiple treatments, with the aim of comparing healthy and diseased samples under the same conditions. OAU1 represents *Zanthoxylum* plants from 103°75'E_31°77'N, OAU2 represents *Panax* plants from 100°3'E_26°49'N, and OAU3 represents *Panax* plants from 103°68'E_24°2'N. Statistical significance was determined using a linear mixed effects model with the formula: relative abundance –health_status + (1|OAU), followed by ANOVA ($n = 11$ biologically independent samples per group).

16S rRNA sequence of *F. anhuiense* K5, was also enriched in the treated soils (Fig. 5C).

The changes in the transcriptome of *F. anhuiense* K5 were determined under 1 mM tocopherol acetate treatment. Tocopherol acetate altered the *F. anhuiense* K5 transcriptome (Supplementary Fig. 11A). There were 187 and 196 genes whose expression was significantly upregulated and downregulated, respectively (Supplementary Fig. 11B). Pathways related to carbohydrate metabolism (C5-branched dibasic acid metabolism; alanine, aspartate and glutamate metabolism; beta-alanine metabolism; and pyruvate metabolism) were enriched in the tocopherol acetate treatment group (Fig. 5D). The expression of the dominant genes TonB-dependent receptor (Supplementary Table 11) and major facilitator superfamily transporter (MFS transporter) (Supplementary Table 12), which are related to nutrient intake, was upregulated. Specifically, the relative expression level of the MFS transporter gene was most strongly upregulated (approximately 10.54-fold) compared with that of the control. Results of qRT–PCR also confirmed that these genes were significantly upregulated compared with those in the control (Fig. 5E, F). However, genes related to gliding motility (Supplementary Table 13) and alpha-amylase (Supplementary Table 14) were downregulated.

### Effect of *F. anhuiense* and tocopherol acetate on the *in planta Fusarium* wilt

We investigated the effects of *F. anhuiense* K5 and tocopherol acetate, both alone and in combination, on tomato *in planta Fusarium* wilt, with assessments conducted at 15 days post-inoculation. The results demonstrated that the control group exhibited the highest disease index, reaching 81.8% (Fig. 6). Treatments with *F. anhuiense* K5 or tocopherol acetate alone significantly reduced the disease index to 39.5% and 38.4%, respectively, approximately half that of the control. The combined application of *F. anhuiense* K5 and tocopherol acetate resulted in the lowest disease index, recorded at only 22.7%. The control efficacy of *F. anhuiense* K5, tocopherol acetate, and the combined treatment was 51.7%, 52.9%, and 72.3%, respectively (Supplementary Table 15). Pathogen density quantification in the rhizosphere corroborated the disease index findings: the control group supported the highest pathogen load ($1.9 \times 10^5$ CFU/mL), whereas the combined treatment group showed the lowest ($6.1 \times 10^4$ CFU/mL). Furthermore, tocopherol acetate significantly increased the abundance of *F. anhuiense* K5, with the bacterial population in the combined treatment showing a 3.16-fold increase compared to the treatment with *F. anhuiense* K5 alone.

### Discussion

Through the re-analysis of published studies related to the *Fusarium* infection microbiome, we firmly established that *Fusarium* infection significantly influences the composition of the bacterial community in the rhizosphere. In particular, the antagonistic *Flavobacterium* (Fig. 3) and antioxidant functions related to sulfur metabolism (Table 1) were significantly increased in the diseased rhizosphere. This general information provides key insights into the assembly mechanism of the microbiome in the *Fusarium* wilt rhizosphere.

*Fusarium* infection could significantly affect bacterial alpha diversities in our study (Supplementary Fig. 6). However, the impact of *Fusarium* infection on bacterial alpha diversities remains complex, with studies reporting both increases and decreases in the diversities[21,22]. These contrasting effects likely depend on environmental conditions or native soil microbial composition. For instance, in low-pH soils, *Fusarium* may produce bikaverin, a metabolite that selectively inhibits certain bacteria (e.g., *Bacillus* and *Actinomycetes*), potentially reducing diversity[23]. Conversely, when a pathogen's activity

**Table 2 | Significantly co-enriched genes in the healthy rhizosphere from the metagenome analysis of different OAUs**

| KO | Function | Dis | Hea | P | Fold (D/H) | OAU |
|---|---|---|---|---|---|---|
| K00029 | malate dehydrogenase | 190.4 ± 23.19 | 259.09 ± 38.5 | 0.0067 | 0.73 | OAU1 |
| | | 221.48 ± 73.87 | 342.42 ± 84.54 | | 0.65 | OAU2 |
| K00249 | acyl-CoA dehydrogenase | 931.63 ± 53.51 | 1156.43 ± 136.45 | 0.0058 | 0.81 | OAU1 |
| | | 906.58 ± 221.48 | 1315.06 ± 314.93 | | 0.69 | OAU2 |
| K00574 | cyclopropane-fatty-acyl-phospholipid synthase | 226.2 ± 24.32 | 281.58 ± 43.13 | 0.0144 | 0.80 | OAU1 |
| | | 212.12 ± 53.97 | 361.39 ± 121.88 | | 0.59 | OAU2 |
| K00831 | phosphoserine aminotransferase | 144.14 ± 15.74 | 200.41 ± 21.98 | 0.0057 | 0.72 | OAU1 |
| | | 113.48 ± 22.23 | 214.71 ± 96.72 | | 0.53 | OAU2 |
| K01251 | adenosyl homocysteinase | 365.79 ± 46.24 | 458.21 ± 22.91 | 0.0008 | 0.80 | OAU1 |
| | | 326.31 ± 64.83 | 418.33 ± 19.27 | | 0.78 | OAU2 |
| K01626 | 3-deoxy-7-phosphoheptulonate synthase | 172.12 ± 18.58 | 229.06 ± 31.54 | 0.0165 | 0.75 | OAU1 |
| | | 140.16 ± 29.3 | 254.93 ± 126.22 | | 0.55 | OAU2 |
| K01679 | fumarate hydratase | 255.62 ± 35.69 | 311.71 ± 26.48 | 0.0309 | 0.82 | OAU1 |
| | | 193.27 ± 44.71 | 221.32 ± 38.4 | | 0.87 | OAU3 |
| K01687 | dihydroxy-acid dehydratase | 456.26 ± 45.12 | 567.64 ± 50.14 | 0.0029 | 0.80 | OAU1 |
| | | 481.64 ± 113.09 | 665.61 ± 97.06 | | 0.72 | OAU2 |
| K03781 | catalase | 175.49 ± 30.82 | 264.62 ± 36.1 | 0.0131 | 0.66 | OAU1 |
| | | 131.53 ± 37.74 | 252.4 ± 149.82 | | 0.52 | OAU2 |
| K03782 | catalase-peroxidase | 168.02 ± 18.03 | 238.03 ± 40.82 | 0.0057 | 0.71 | OAU1 |
| | | 145.62 ± 25.76 | 168.08 ± 15.5 | | 0.87 | OAU3 |
| K07716 | two-component system, cell cycle sensor histidine kinase | 147.58 ± 30.2 | 191.43 ± 9.25 | 0.0039 | 0.77 | OAU1 |
| | | 141.84 ± 57.22 | 232.44 ± 36.45 | | 0.61 | OAU2 |
| K21470 | L,D-transpeptidase | 169.25 ± 8.05 | 209.44 ± 17.3 | 0.0032 | 0.81 | OAU1 |
| | | 109.52 ± 50.95 | 198.49 ± 40.45 | | 0.55 | OAU2 |
| K03694 | ATP-dependent Clp protease ATP-binding subunit | 243.38 ± 34.79 | 335.67 ± 32.12 | 0.0013 | 0.73 | OAU1 |
| | | 219.41 ± 66.41 | 313.64 ± 51.29 | | 0.70 | OAU2 |

OAU1 represents *Zanthoxylum* plants from 103°75′E_31°77′N, OAU2 represents *Panax* plants from 100°3′E_ 26°49′N, and OAU3 represents *Panax* plants from 103°68′E_ 24°2′N. Statistical significance was determined using a linear mixed effects model with the formula: relative abundance ~ health_status + (1|OAU), followed by ANOVA (*n* = 11 biologically independent samples per group).

involves primarily saprophytic functions (e.g., lignin degradation) without antibacterial effects, it may indirectly promote bacterial fitness or diversity by releasing supplemental nutrient resources[24].

A previous microbiome meta-analysis suggests that the majority of rhizosphere microbial taxa were affected by *Fusarium* infection[25]. However, knowledge related to microbiome functions is still generally lacking. In the present study, we reanalyzed 22 metagenomic samples to reveal changes in microbiome functions. The gene encoding acetylhexosaminidase, which is related to the degradation of chitin, was significantly enriched in the diseased rhizosphere (Table 1). These results suggest that certain beneficial bacteria that can inhibit the growth of fungi are enriched in diseased rhizospheres, which may contribute to the development of the disease-suppressive soil[7,26]. In addition, we confirmed that the gene cassettes involved in antioxidant functions related to sulfur metabolism and catalase were significantly enriched in the diseased and healthy rhizospheres, respectively. Thioredoxin peroxidase is an important peroxidase that can protect organisms against stressful environments[27]. Compared with catalase, thioredoxin peroxidase has a wider antioxidant range[28,29]. These results suggest that the different oxidation products were enriched in the diseased rhizosphere. Yuan et al. reported that certain beneficial bacteria of *Bacillus* and *Sphingomonas* can release reactive oxygen species to inhibit the growth of fungal pathogens in the rhizosphere of *F. oxysporum* inoculated plants[7]. Moreover, diseased plants can release reactive oxygen species to protect themselves from pathogen infection[30]. This information suggests that the microbiome may suffer from reactive oxygen species stress. This environment may enhance the ecological niche of bacteria, which have a high antioxidant capacity. In addition, two genes encoding thioredoxin peroxidase were

found in the *F. anhuiense* K5 genome, which may contribute to adaptation to such an environment (Supplementary Table 8).

Identifying the general changes in the bacterial composition of the *Fusarium* wilt rhizosphere may lead to improvements in the control of soil-borne pathogens by bacteria that adapt well to the rhizosphere. We found that enrichment of the antagonistic bacteria *Flavobacterium* was a general signature in the diseased rhizosphere (Fig. 2B). *Flavobacterium* is significantly enriched in resistant tomato cultivars and can protect the host from *Ralstonia solanacearum* infection[31]. In addition, beneficial *Flavobacterium* are highly enriched in endophytic roots and depress the growth of the pathogen *Rhizoctonia solani*[5]. We found that the genome of *F. anhuiense* K5 contains genes related to the degradation of chitin, which may help strain K5 to inhibit the growth of fungal pathogens (Supplementary Table 8). Overall, these results suggest that the enrichment of beneficial *Flavobacterium* in the rhizosphere is a stable feature in the infected plants regardless of the plant, pathogen or soil type. This information provides hope for the stable control of soil-borne diseases.

Root exudates are the main drivers of the assembly of the rhizosphere microbiome. Although many previous studies have demonstrated the changes in the components of the root exudates of diseased plants[15,32,33], the components that can stimulate *Flavobacterium* growth are still unknown. Here, we investigated the enriched components in the root exudates of diseased plants. We found that the root secreted tocopherol acetate was significantly enriched in the diseased rhizosphere. It is likely that diseased plants may suffer from oxygen stress, leading to the release of tocopherol acetate[34]. Moreover, tocopherol acetate significantly stimulated the growth of *F. anhuiense* K5 (Fig. 4). A previous study reported that the relative

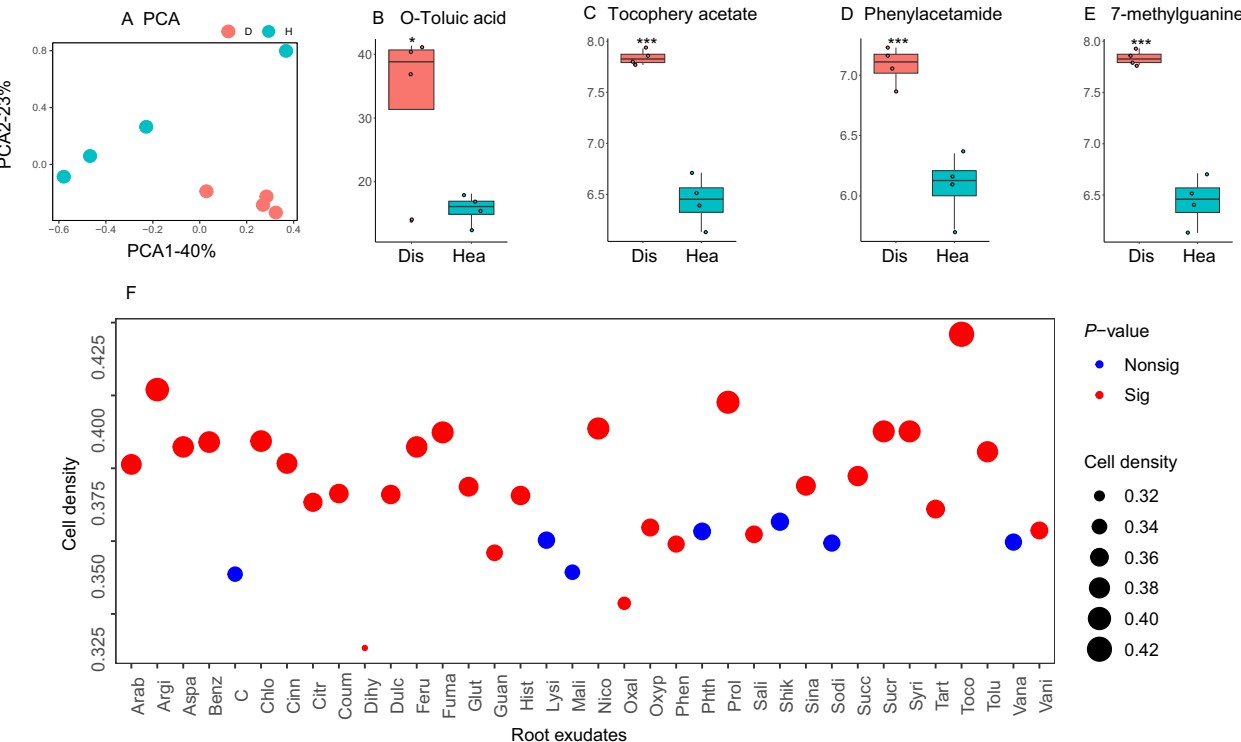

**Fig. 4 | *Fusarium* infection affected the tomato root exudate composition.**
**A** Principal component analysis of tomato root exudates. **B**–**E** Shown are the four most abundant differentially expressed metabolites. Data are presented as box plots (center line, median; box limits, upper and lower quartiles (25th/75th percentiles); whiskers, min/max). Each data point represents an independent biological sample ($n = 4$ biologically independent samples). Statistical significance was determined using a Student's two-sided $t$ test. Exact $P$ values: o-toluic acid $= 3.00 \times 10^{-2}$, tocopherol acetate $= 3.27 \times 10^{-5}$, phenylacetamide $= 6.18 \times 10^{-4}$, 7-methylguanine $= 3.34 \times 10^{-5}$. **F** The effects of 35 enriched root exudates on the growth of *F. anhuiense* K5 at 19 h (stationary phase). Circle size represents the cell density ($OD_{600}$). Red and blue colors indicate components that significantly ($P < 0.05$) or non-significantly altered bacterial growth, respectively (Student's

two-sided $t$ test, $n = 3$ biological replicates per condition). ***$P < 0.001$, **$P < 0.01$, *$P < 0.05$. Metabolite abbreviations: Arab (arabitol), Argi (arginine), Aspa (aspartic acid), Benz (benzoic acid), Chlo (chlorogenic acid), Cinn (cinnamic acid), Citr (citric acid), Coum (coumaric acid), Dulc (dulcitol), Feru (ferulic acid), Fuma (fumaric acid), Glut (glutamic acid), Guan (7-methylguanine), Hist (histidine), Hydr (p-hydroxybenzoic acid), Lysi (L-lysine), Mali (malic acid), Nico (nicotinic acid), Oxal (oxalic acid), Oxyp (4-methoxypyridine), Phen (2-phenylacetamide), Phth (phthalic acid), Prol (L-proline), Sali (salicylic acid), Shik (shikimic acid), Sina (sinapic acid), Sodi (sodium glycolate), Succ (succinic acid), Sucr (sucrose), Syri (syringic acid), Tart (tartaric acid), Toco (tocopherol acetate), Tolu (O-toluic acid), Vana (vanillic acid), and Vani (vanillin). Source data are provided as a Source Data file.

abundance of *Flavobacterium* was significantly increased in soil supplemented with tocopherol acetate, citrulline, galactitol, and behenic acid[15]. In the present study, genes associated with carbohydrate metabolism were significantly upregulated in *F. anhuiense* K5 following tocopherol acetate treatment. (Fig. 5D). The enrichment of *F. anhuiense* K5 by tocopherol acetate was also demonstrated by the *in planta* assay (Fig. 6). The combined application of *F. anhuiense* K5 and tocopherol acetate resulted in the lowest disease index among all treatments. It is postulated that the underlying mechanism for the enhanced efficacy involves both the enrichment of *F. anhuiense* K5 by tocopherol acetate and the concomitant tocopherol-mediated activation of plant immune responses[35,36]. Collectively, our findings suggest that the combined application of *F. anhuiense* K5 and tocopherol acetate offers a potential strategy for managing tomato *Fusarium* wilt in a hydroponic-based system. Future research will focus on the development of specific control measures, including evaluating the long-term durability of the protection and its effectiveness in complex soil environments, which are beyond the objectives of this study.

In conclusion, our results indicate that the beneficial bacterium *Flavobacterium* is a general signature of the *Fusarium* wilt rhizosphere, which may lead to enhanced control of soil-borne pathogens by bacteria that adapt well to the rhizosphere. Moreover, the antioxidant functions related to sulfur metabolism and the root secreted tocopherol acetate were strongly enriched in the diseased rhizosphere,

which provides a basis for understanding the bacterial assembly mechanism in the *Fusarium* wilt rhizosphere (Fig. 6D).

## Methods
### Study selection
The following search terms were entered into PubMed, Google Scholar, and the NCBI Sequence Read Archive (SRA) to obtain an unbiased representation of studies on the effects of *Fusarium* infections on the rhizosphere bacterial community: *Fusarium* disease & rhizosphere, *Fusarium* rhizosphere microbiome, *Fusarium* infection & plant microbiota, *Fusarium* & plant disease microbiota, *Fusarium* rot root & rhizosphere and soil-borne diseases. The search yielded 37 and 4 potential amplicon and metagenomic studies based on the inclusion criteria, respectively (Supplementary Table 1). We checked the title, abstract, year, and journal in these studies. For a study to be included in the meta-analysis, it had to include amplicon sequencing data and a detailed sequencing analysis method for studying the effects of *Fusarium* infection on the rhizosphere microbiome. In addition, the studies include at least six samples, each with available metadata and sequencing data. The definitions of diseased and healthy samples were based on the treatments indicated in published studies.

Within a single study from our meta-dataset, variations in sampling locations and times can significantly disrupt the comparison

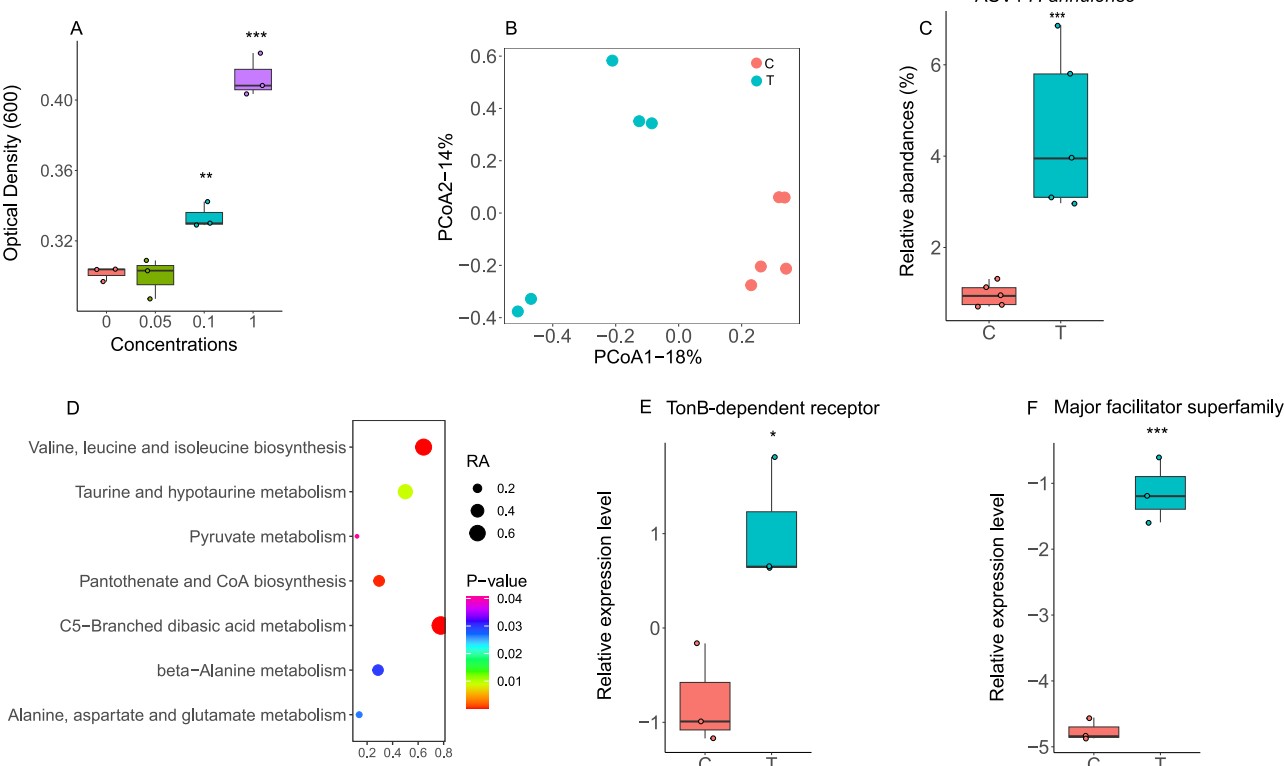

**Fig. 5 | Tocopherol acetate-enriched *F. anhuiense* K5. A** Growth of *F. anhuiense* K5 after treatment with different concentrations of tocopherol acetate for 12 h in vitro (*n* = 3 biologically independent samples per group). Compared with the control, the *P* values for the 0.1 mM and 1 mM groups were $2.72 \times 10^{-3}$ and $3.00 \times 10^{-4}$, respectively (two-sided Student's *t* test). **B** Principal coordinate analysis of the bacterial community in soil amended with tocopherol acetate (*n* = 5 biologically independent samples per group). **C** Enrichment of *F. anhuiense* K5 (ASV1) in soil amended with 1 mmol/kg tocopherol acetate (*n* = 5 biologically independent samples per group). The exact FDR-adjusted *P* value for ASV1, determined using the DESeq2 package (negative-binomial Wald test with Benjamini-Hochberg FDR correction), was $7.07 \times 10^{-10}$. **D** Enriched pathways of *F. anhuiense* K5 induced by 1 mM

tocopherol acetate (*n* = 3 biologically independent samples per group). Pathway enrichment was assessed using a hypergeometric test. The significantly enriched pathways (*P* < 0.05) are shown. **E, F** Relative expression levels of genes related to the TonB-dependent receptor (**E**) and the major facilitator superfamily transporter (**F**) in *F. anhuiense* K5 under control and tocopherol acetate treatment (*n* = 3 biologically independent samples per group). Compared with the control, the *P* values were $2.22 \times 10^{-2}$ and $2.87 \times 10^{-4}$, respectively (two-sided Student's *t* test). ****P* < 0.001, ***P* < 0.01, **P* < 0.05. Data in **A**, **C**, **E**, and **F** are presented as box plots showing the median (center line), the 25th and 75th percentiles (box limits), and the minimum and maximum values (whiskers). Source data are provided as a Source Data file.

between healthy and diseased rhizosphere microbiomes. To address this, we introduced the concept of OAU, which refines the analysis by subdividing samples into subgroups until comparisons between diseased and healthy rhizosphere microbiomes can be made under the same conditions (e.g., same sampling location and time within a study). Samples from the same sampling location or time are grouped into one OAU. These subgroups are defined as OAUs. (Supplementary Fig. 12).

**Amplicon sequencing data analysis**
The amplicon sequencing data were downloaded from the SRA. Sequence adapters were removed via Trim Galore v0.6.10 (https://github.com/FelixKrueger/TrimGalore), and sequence quality was tested via FastQC v0.12.1 (https://www.bioinformatics.babraham.ac.uk/projects/fastqc/). The base sequences with a Phred quality score below 30 were removed. Because 90-bp sequences were sufficiently long to reveal detailed patterns of community structure, only the forward reads were considered and were cut to 100 bp[37]. The reads were filtered based on quality (where available) again via vsearch[38] with the following parameters: --fastq_maxee_rate 1.0. Unique sequences without chimeras were obtained with the following functions: vsearch --derep_fulllength and vsearch --uchime3_denovo. Finally, ASVs were selected via vsearch -- cluster_unoise and were annotated against the RDP database at a cutoff of 0.8. The abundances of ASVs were obtained by vsearch --usearch_global.

**Metagenomic sequencing data analysis**
The metagenomic sequencing data downloaded from the SRA were preprocessed by Fastp v 0.23.4 to trim the adapters and remove the low-quality (*q* < 15), short (<15 reads) and low-complexity sequences[39]. Bowtie2 v 2.5.1 was used to map the reads to the host genome, and finally, the unmapped reads were reserved as clean sequences[40]. MEGAHIT v1.29 was used to assemble the clean sequences[41]. The assembled sequences were predicted for open reading frames via Prodigal v2.6.3[42]. CD-HIT v4.8.1 was applied to cluster the sequences at a similarity threshold of 0.95, resulting in a nonredundant gene set[43]. EggNOG mapper v1.0.3 was used to annotate these genes on the basis of the eggNOG database[44]. The annotation results were further integrated and categorized on the basis of orthologous genes (KOs) within the KEGG database. Salmon v1.10.2 was used to quantify the gene abundances[45].

**Diversity analysis**
The amplicon samples with a sequence depth of <5000 were removed. The remaining samples were rarefied to the minimum sequence depth to ensure consistent analysis. The estimate_richness function of the phyloseq v 1.41.1 package was used to generate the Shannon, observed richness, ACE, Chao, Simpson and Faith phylogenetic diversity indices[46]. The combined analysis, which was based on a previous study[47] was conducted via a linear mixed effects model via the lmer function of lme4 v1.1.34[48] with the formula alpha diversity index

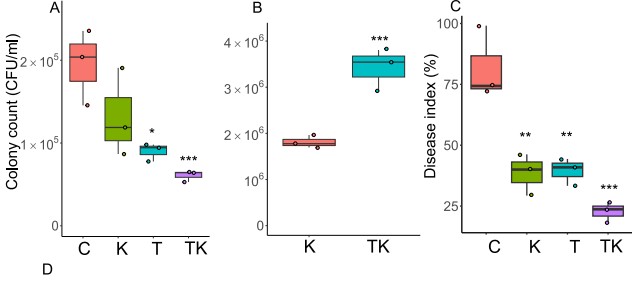

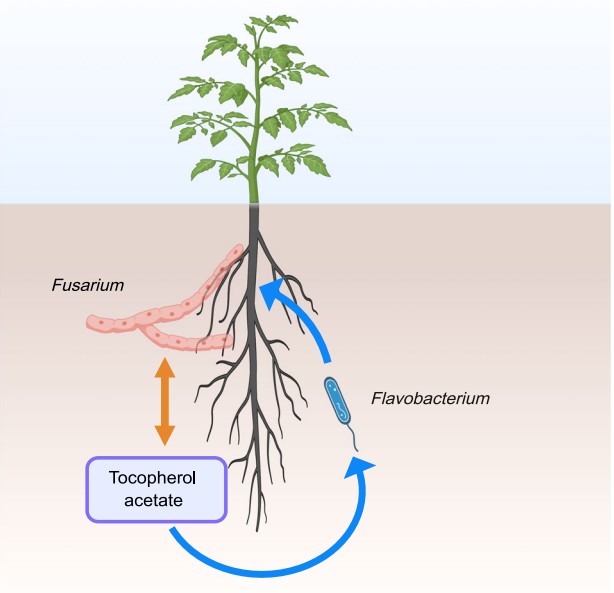

**Fig. 6 | Inhibitory effect of *F. anhuiense* against *F. oxysporum* and *F. anhuiense* population enhancement by tocopherol acetate *in planta*.** *In planta* experiments (*n* = 3 biologically independent samples per group; each sample consisted of a pooled collection from 30 tomato plants). Treatments: C (control), T (tocopherol acetate), K (F. anhuiense K5), TK (tocopherol acetate+F. anhuiense K5). Spores of *F. oxysporum* were inoculated in all treatments. **A** Colony count of *F. oxysporum* at 15 days. Compared with the control, the *P* values for the K and TK groups were $1.76 \times 10^{-2}$ and $4.50 \times 10^{-3}$, respectively (one-way ANOVA followed by Dunnett's post hoc test). **B** Colony counts of *F. anhuiense* K5 at 15 days. Compared with the control, the *P* value for the K group was $4.69 \times 10^{-3}$ (Student's two-sided *t* test). **C** Disease index at 15 days. Compared with the control, the *P* values for the K, T and TK groups were $1.24 \times 10^{-3}$, $1.30 \times 10^{-3}$, and $1.60 \times 10^{-4}$, respectively (one-way ANOVA followed by Dunnett's post-hoc test). Data in **A**–**C** are presented as box plots showing the median (center line), the 25th and 75th percentiles (box limits), and the minimum and maximum values (whiskers). **D** Proposed model for *Flavobacterium* enrichment in the rhizosphere of infected plants. Pathogen infection enhances root secretion of tocopherol acetate, which promotes beneficial *Flavobacterium* growth. The recruited *Flavobacterium* may subsequently suppress *F. oxysporum* invasion through antimicrobial activity. Orange and blue arrows represent root-exudate–pathogen and root-exudate–probiotic interactions, respectively. Created in BioRender. Lv, Su. (2025) https://www.biorender.com/7cods4t. Source data are provided as a Source Data file.

-healthy or diseased treatment + (1 | OAU), and significance was determined via the ANOVA function. Beta-diversity was assessed by computing: (1) Bray-Curtis dissimilarity based on genus-level abundance profiles, and (2) weighted UniFrac distance matrices derived from ASV-level phylogenetic data, using the vegan v 2.6.4[43] and ape v 5.7.1[49] package in R. To determine the significance in difference of community composition between healthy and diseased samples, a similarity analysis (ANOSIM) was performed using the anosim function

in vegan package v 2.6.4[50]. The homogeneity of dispersion between healthy and diseased samples was tested using the betadisper function in the vegan package v 2.6.4[50]. The combined analysis of community structure was similar to that of alpha diversity, with the following formula: PcoA1 value - healthy or disease treatment + (1 | OAU).

## Effects of *F. oxysporum* invasion on the tomato and *Panax notoginseng* rhizosphere bacterial communities

The greenhouse experiment included one treatment group and one control group, with the following design: (i) control, without the tomato pathogen *Fusarium oxysporum*; and (ii) treatment, with application of the pathogen *F. oxysporum*. Sixty seedlings (four-leaf stage) from the treatment and control groups were planted in 60 plastic plant pots containing ~0.5 kg of soil per pot (30 plants for the control group and 30 plants for the treatment group). The soil was collected from a vegetable production center in the urban area of Nanjing, China (118°57′E, 32°03′N). The soils used were luvisol (FAO) soils. The soil chemical properties are shown in Supplementary Table 16. The pot experiment was performed in a greenhouse (day temperature: 28 °C; night temperature: 25 °C). Fifteen days after transplantation, 20 mL of pathogen suspension (~$1 \times 10^6$ CFU mL⁻¹) and sterile water were poured into the soil near the root shoot in the treatment and control group. After 33 days, 15 plants from the treatment group and seven plants from the control group were randomly selected, and the rhizosphere soils were collected. The harvested tomato roots were vigorously shaken to remove the loosely bound soil and collected into 50-mL Falcon tubes with 15 mL of sterile PBS solution. The wash buffer was subjected to centrifugation ($1500 \times g$, 20 min), and the resulting pellet was defined as the rhizosphere soil. Once the rhizosphere soil samples were obtained, they were immediately subjected to soil DNA extraction. In total, 22 rhizosphere soil samples were obtained for soil DNA extraction.

The diseased and healthy *P. notoginseng* plants were obtained from the *P. notoginseng* plantation in Wenshan County of Yunnan Province (23°34′N, 104°19′E, 1500 m alt.), a geo-authentic production area for *Panax notoginseng* in China. As Wang et al.[51] described, information regarding the soil type and climate in this region has been previously reported. A two-hectare *P. notoginseng* plantation was created in October 2016, and seeds were sown in December 2023. When the incidence of root rot in the plantation exceeded 10% in the second year of *P. notoginseng* plant growth, soil sampling started. Rhizosphere soil samples were collected from healthy and diseased *P. notoginseng* plants from the plantation. Plants with firm stems, green leaves, and roots lacking rust spots, rot, or other disease symptoms were classified as healthy. In contrast, plants with yellowing, wilting, and root rot were classified as severely diseased[52]. Ten diseased and healthy plants were randomly collected. The root tissue with rhizosphere soil was transported to the laboratory at 4 °C. The process of collecting the rhizosphere soil was similar to that of collecting the tomato rhizosphere soil. In total, 20 rhizosphere soil samples were obtained for soil DNA extraction.

## Isolation of *Flavobacterium* in the tomato rhizosphere

Five tomato seedlings were planted in the soil from Nanjing as described above. The tomato planting process was similar to the previous greenhouse experiment, with each pot containing 0.5 kilograms of soil. A 20 mL pathogen suspension (~$1 \times 10^6$ CFU mL⁻¹) was inoculated into the root-stem junction of 30-day-old seedlings. After 10 days, the freshly sampled rhizosphere soils of tomato were serially diluted on TSA and PSR2A-C/T (selective media for *Flavobacterium*)[53] agar plates. The plates were incubated for 3–5 days at 30 °C. Yellow colonies (36) were selected. Sanger sequencing of the 16S rRNA gene was performed with 27 F and 1492 R as the sequencing primers by Qin Ke Company (Nanjing, China).

### Draft genomes of *F. anhuiense* K5

Genomic DNA was extracted by using the cetyltrimethylammonium bromide method with minor modifications, and then the DNA concentration, quality and integrity were determined by using a Qubit Flurometer (Invitrogen, USA) and a NanoDrop Spectrophotometer (Thermo Scientific, USA). Sequencing libraries were generated using the TruSeq DNA Sample Preparation Kit (Illumina, USA) and the Template Prep Kit (PacificBiosciences, USA). Genome sequencing was then performed by Personal Biotechnology Company (Shanghai, China) using the Illumina Novaseq platform. The low-quality sequences were removed via adapter removal v2.1.7[54]. The high-quality paired-end reads were quality corrected by SOAPec v2.0 on the basis of the k-mer frequency, with the k-mer used for correction set to 17[55]. The genome was assembled de novo via A5-miseq v20,150,522[56]. The coding DNA sequences in the draft genomes were predicted via GeneMarkS v4.32[57]. The predicted CDs were searched against the NCBI NR protein database and the KEGG database. The whole-genome shotgun project has been deposited in GenBank under the accession number PRJNA1138398.

### In vitro inhibition of *Fusarium* by *F. anhuiense* K5

*F. anhuiense* K5 was inoculated in TSB liquid medium and incubated at 30 °C with 180 rpm for 24 h. Two hundred microliters of a strain K5 cell suspension (cell density = $10^8$ cfu/mL) was spotted onto the side of PDA medium agar plates. The agar plugs of the tomato pathogen *Fusarium oxysporum* were placed on the other side of the PDA medium agar plates. The inoculated PDA media agar plates containing pathogens and strain K5 were incubated at 28 °C for 5 days.

### Effects of root exudates on the growth of *F. anhuiense* K5

The collection of tomato root exudates was conducted according to published studies[58,59]. Fifty-six tomato seedlings (four-leaf stage) were planted in 56 plastic plant pots containing ~0.5 kg of soil per pot (28 plants for the control group and 28 plants for the treatment group). Four replicates were used, and each replicate comprised seven plants. Fifteen days after transplantation, 20 mL of pathogen suspension (~$1 \times 10^6$ CFU mL$^{-1}$) and sterile water were poured into the soil near the root shoot in the treatment and control group. Root metabolites from the control and treatment group tomato plants were collected from the pot experiment. After 10 days, the root soil was washed with sterilized deionized water. After a recovery period of 3 days, the roots were transferred to root metabolite collection flasks for hydroponic analysis. The water was changed once a day during the recovery period. After 24 h of incubation in a greenhouse, the sediment was weighed after freeze-drying. A total of eight mass spectrometry samples were analyzed, comprising treatment and control groups, each with four independent biological replicates. Quality Control (QC) samples were prepared by pooling equal aliquots from all experimental samples to monitor instrument stability during the sequence. Approximately 25 mg of frozen samples was weighed and extracted with 1 mL of pre-cooled methanol: water (3:1, v/v). Homogenization was performed using a grinder (35 Hz, 4 min) followed by ice-water bath ultrasonication (5 min, repeated 3 times). The extract was kept at −40°C for 1 h and then centrifuged at 12,000 rpm (RCF = 13,800 × g, radius = 8.6 cm) for 15 min at 4 °C. A 100 μL aliquot of the supernatant was transferred and dried in a vacuum concentrator. Derivatization was performed in two steps: first, 30 μL of methoxyamine hydrochloride in pyridine (20 mg/mL) was added and incubated at 80 °C for 30 min; then, 40 μL of BSTFA (with 1% TMCS, v/v) was added and the mixture was heated at 70 °C for 1.5 h. After cooling to room temperature, samples were subjected to GC-TOF-MS analysis. Metabolite profiling was conducted using an Agilent 7890 gas chromatograph coupled with a time-of-flight mass spectrometer. Separation was achieved on a DB-5MS capillary column. Helium was used as the carrier gas at a constant flow rate of 1 mL/min. A 1 μL sample was injected in splitless mode. The oven temperature was programmed as follows: held at 50°C for 1 min, ramped at 10 °C/min to 310 °C, and held for 8 min. The injector, transfer line, and ion source temperatures were set to 280 °C, 280 °C, and 250 °C, respectively. Electron impact ionization was employed at −70 eV. Mass spectra were acquired in full-scan mode from m/z 50 to 500 at an acquisition rate of 12.5 spectra per second, with a solvent delay of 6.25 min. Raw data were processed using Chroma TOF software (version 4.3x, LECO) for peak picking, baseline correction, deconvolution, alignment, and integration. Metabolites were identified by matching mass spectra and retention indices against the LECO-Fiehn Rtx5 database. Rigorous QC was applied: features detected in <50% of QC samples or with a relative standard deviation (RSD)>30% in QC samples were removed.

### Identification of root exudates in *Fusarium*-infected plants from published studies

The search terms (*Fusarium* and root exudates) were entered into PubMed and Google Scholar to obtain an unbiased representation of studies. The studies that included changes in the abundances of root exudates related to *Fusarium* infections were reviewed. The definitions of diseased and healthy samples were based on the treatments indicated in published studies.

### Effects of root exudates on the growth of *F. anhuiense* K5

The *F. anhuiense* K5 strain was cultured in 200 mL of tryptic soy broth (TSB) liquid medium at 30 °C for 24 h at 170 rpm. The culture was centrifuged at 10,000 × g/min. The supernatant was removed. The strain was washed with 1/10 TSB, and the OD value was adjusted to 1. Then, 2 μl of culture was transferred to 200 μl of 1/10 TSB medium containing the root exudate component at a concentration of 1 mM in 96-well plates on the basis of a previous study[60]. The 35 root exudates included metabolites Arab (arabitol), Argi (arginine), Aspa (aspartic acid), Benz (benzoic acid), Chlo (chlorogenic acid), Cinn (cinnamic acid), Citr (citric acid), Coum (coumaric acid), Dulc (dulcitol), Feru (ferulic acid), Fuma (fumaric acid), Glut (glutamic acid), Guan (7-methylguanine), Hist (histidine), Hydr (p-hydroxybenzoic acid), Lysi (L-lysine), Mali (malic acid), Nico (nicotinic acid), Oxal (oxalic acid), Oxyp (4-methoxypyridine), Phen (2-phenylacetamide), Phth (phthalic acid), Prol (L-proline), Sali (salicylic acid), Shik (shikimic acid), Sina (sinapic acid), Sodi (sodium glycolate), Succ (succinic acid), Sucr (sucrose), Syri (syringic acid), Tart (tartaric acid), Toco (tocopherol acetate), Tolu (O-toluic acid), Vana (vanillic acid), and Vani (vanillin). A culture without the root exudate component was used as a control. Each treatment has three replicates. The cultures were then incubated at 30 °C with shaking for 48 h, after which the $OD_{600}$ of each culture was monitored every hour by a microbial growth curve analyzer. To further determine the effects of tocopherol acetate on the growth of *F. anhuiense* K5 at different concentrations and times, a process similar to that described above was performed. However, the growth of *F. anhuiense* K5 at concentrations of 0.05 mM, 0.1 mM, and 1 mM was determined at 12 h.

### Effects of tocopherol acetate on the soil bacterial community

This experiment includes a treatment (soil amended with tocopherol acetate) and a control. Approximately 15 mL of sterilized water was added to 300 g of the soil described above to wet the soil. Next, 20 mL of an *F. anhuiense* K5 cell suspension was added to the wet soil and mixed (final cell concentration: $10^8$ cfu g$^{-1}$ soil). The soil was then divided evenly into two parts. The soils treated with or without tocopherol acetate were the treatment and control, respectively. The treatment was applied at a concentration of 1 mmol/kg in the soil. The soil moisture of both the treatment and the control was adjusted to 45% of the soil water-holding capacity. Then, 150 g of soil was divided into five replicates consisting of 30 g of soil in a 50-mL centrifuge tube

and incubated at 28 °C. The soils were randomly taken from the treatment and control groups on day 5 for soil DNA extraction for high-throughput sequencing.

### *In planta* assay of the effects of *F. anhuiense* and tocopherol acetate on *F. oxysporum*

The experimental design comprises three treatments: (1) tocopherol acetate application, (2) *F. anhuiense* K5 inoculation, and (3) combined tocopherol acetate and *F. anhuiense* K5 application, along with an untreated control. All treatments and controls were inoculated with *F. oxysporum* spores. The methodology followed established protocols from the previous study[61] with modifications. Tomato seeds were surface-sterilized in 2% (v/v) sodium hypochlorite solution for 15 minutes, followed by five rinses with sterile distilled water to remove residual disinfectant. About thirty sterilized seeds were aseptically placed on sterile nylon mesh (0.5 mm pore size) supported by four vertically oriented 200-μL pipette tips in a hydroponic culture system. The assembly was maintained in sterile culture vessels containing 200 mL of half-strength Hoagland's nutrient solution, with the solution level carefully adjusted to maintain continuous mesh contact without submergence. Plants were cultivated in a controlled-environment growth chamber under 16/8 h light/dark cycles at 28 °C (day) and 21 °C (night) for 20 days[62]. Sterile stock solutions were prepared for treatment applications: tocopherol acetate (1 mM final concentration in hydroponic system), *F. anhuiense* K5 cell suspension ($1 \times 10^6$ CFU mL$^{-1}$)[63], and *F. oxysporum* spore suspension ($1 \times 10^5$ CFU mL$^{-1}$)[64]. The experiment was conducted with three biological replicates per treatment. Each replicate consists of ~30 seedlings. Fifteen days after treatment, 15 mL of hydroponic solution was collected from each replicate. Disease severity was evaluated using a 0–4 rating scale, where 0 indicates a complete absence of infection, and 4 represents complete plant infection[62]. The detailed criteria for the disease severity scale are as follows: 0 - No infection: Plants show no visible symptoms of disease. 1 - Mild infection (approximately 25% severity): One to two leaves exhibit yellowing symptoms. 2 - Moderate infection (approximately 50% severity): Two to three leaves show yellowing, with about half of the leaves displaying wilting symptoms. 3 - Severe infection (approximately 75% severity): All plant leaves become yellow, approximately 75% of leaves show wilting, and plant growth is significantly stunted. 4 - Complete infection (100% severity): All leaves are completely yellow and wilted, leading to plant death. Disease index was calculated using the formula: Disease index (%) = ∑(scale×number of plants infected)/(highest scale×total number of plants)×100. Suppression efficacy of the treatments was determined as follows: Suppression efficacy (%) = (disease index of the control - disease index of treatment)/disease index of the control ×100. DNA was extracted using the FastDNA SPIN Kit for Water (MP Biomedicals, CA) following manufacturer's protocols.

### Amplicon sequencing data analysis in the validation experiments

Total genomic DNA from about 0.5 g of soil samples was extracted using the FastDNA SPIN Kit for Soil (MP Biomedicals, CA). The rhizosphere bacterial 16S rRNA V3-V4 region (F: 5'-ACTCCTACGGGAGG-CAGCA-3'; R: 5'- GGACTACHVGGGTWTCTAAT-3') was sequenced on an Illumina Novaseq 6000 platform (Illumina, San Diego, CA, USA) provided by Personal Biotechnology Co., Ltd. (Shanghai, China). The raw sequence data were filtered via vsearch v2.23[38] software to obtain clean data. The raw reads were analyzed with the fastq_mergepairs, fastx_filter, derep_fulllength, cluster_unoise and uchime3_denovo functions to obtain ASVs with standard settings in the vsearch pipeline. Representative sequences of each ASV were selected and classified by the RDP classifier at a cutoff of 80%. The raw sequence data were deposited in the SRA under accession numbers PRJNA1138403, PRJNA1240447 and PRJNA1240449.

### Transcriptome analysis

Ten milliliters of TSB medium (1/10) from the strain ($OD_{600} = 1.0$) was induced with tocopherol acetate at a concentration of 1 mM in a growth chamber at 30 °C and 170 rpm for 30 min. One milliliter of cells was pelleted by centrifugation at $8000 \times g$ for 2 min and resuspended twice in TSB medium (1/10). The resuspended cells were subsequently frozen in liquid nitrogen, and total RNA was extracted via TRIzol Reagent. The RNA samples were immediately stored at 80 °C for RNA-seq and quantitative RT-PCR.

For RNA-seq, samples were sequenced with a NovaSeq 6000 (Shanghai Personal Biotechnology Cp. Ltd.). The quality and integrity of each sample were determined via a NanoDrop spectrophotometer (Thermo Scientific) and a Bioanalyzer 2100 system (Agilent). A Zymo-Seq RiboFree Total RNA Library Kit was used to remove rRNA from total RNA. Random oligonucleotides and SuperScript III were used to synthesize the first-strand cDNA. Second-strand cDNA synthesis was subsequently performed via DNA polymerase I and RNase H. The remaining overhangs were converted into blunt ends via exonuclease/polymerase activities, and the enzymes were removed.

The high-quality raw RNA-seq data were filtered via fastp (0.22.0) software[39]. The reference genome index was built with Bowtie2 (v2.5.1), and the filtered reads were mapped to the reference genome via Bowtie2[40]. The gene read count value was determined via HTSeq (v0.9.1)[65] and represents the original expression level of the gene. The raw sequence data were deposited in the SRA under accession number PRJNA1138414.

For the real-time quantitative RT–PCR assay, cDNA was synthesized via TransScript One-Step RT–PCR Removal and cDNA Synthesis SuperMix (Takara, Dalian). qRT–PCR was performed via an ABI 7500 real-time PCR system (Applied Biosystems, Waltham, MA, USA). The bacterial qRT–PCR primers (Supplementary Table 17) for several selected genes related to TonB-dependent transporters (TBDTs), the major facilitator superfamily (MFS) and the constitutively expressed normalization gene *gyrA* were synthesized. The PCR protocol consists of an initial denaturation at 95 °C for 10 min, followed by 40 cycles of denaturation at 95 °C for 5 s and annealing/extension at 60 °C for 30 s. The melting curve was systematically checked at the end of each real-time PCR program. The relative expression levels of the target genes were calculated via the 2(−Delta Delta C(T)) method[66]. Each treatment includes three independent replicates.

### Quantification of colony counts of the *in planta* assay

The cycle threshold (Ct) values of *F. oxysporum TEF1* (translation elongation factor 1-α) and *F. anhuiense*-specific sequence fragment were determined by quantitative PCR (qPCR). Primer sequences are provided in Supplementary Table 17. qPCR efficiency ranged from 90 to 105%. To obtain pathologically meaningful data, the pathogen quantification from qPCR result was converted to colony-forming unit (CFU). To establish a standard curve correlating Ct values with colony counts, a starting *F. oxysporum* spore suspension at $2 \times 10^8$ CFU/mL (Supplementary Fig. 13A) and *F. anhuiense* K5 cell suspension at $1.25 \times 10^9$ CFU/mL (Supplementary Fig. 13B) were subjected to serial dilutions, followed by DNA extraction and qPCR analysis for each dilution gradient.

### Statistical analysis

Unless otherwise specified, the statistical analysis was performed in R 4.1.1. Significance was determined according to a *P* value < 0.05. Significant differences in the relative abundances of microbial taxa and expression levels of genes were identified via the DESeq function of the DESeq2 v1.38.3 package[67]. The combined analysis of relative genus abundances was conducted via a linear mixed effects model with the following formula: relative abundance ~ healthy or disease treatment + (1 | OAU). Significance was determined via the ANOVA function.

To identify robust microbial biomarkers distinguishing healthy and diseased plant rhizospheres, we implemented a comprehensive analytical pipeline combining statistical and machine learning approaches. First, we performed initial feature selection using the rfPermute package v2.5.2 (https://github.com/ericarcher/rfpermute) with 999 permutations, analyzing relative abundances of all bacterial genera ($n = 198$ samples) to identify the statistically significant predictors (Type 1 biomarkers) based on statistical changes in MeanDecreaseGini or MeanDecreaseAccuracy index. Then, we selected the top 20 statistically significant predictors based on MeanDecreaseGini or MeanDecreaseAccuracy index (total 31 genera from Type 1 predictors) and optimized model performance through automated hyperparameter tuning using the mlr v2.19.1 framework (https://mlr.mlr-org.com). To enhance reliability, we subsequently calculated SHapley Additive exPlanations (SHAP) values via the shapviz package v0.10.1 (https://github.com/ModelOriented/shapviz) and cross-validated these with the model's built-in importance measures.

The following data followed a normal distribution according to the Shapiro–Wilk test. The cell density ($OD_{600}$) of the *F. anhuiense* K5 strain in vitro, the abundances of the root exudates, the bacterial diversity index values, and the expression levels of genes related to TBCTs and the major facilitator superfamily were normally distributed and assessed via the two-tailed Student's *t* test. The numbers of *F. oxysporum* and *F. anhuiense* exhibited a normal distribution with homogeneity of variance. Significant differences between groups were assessed using one-way ANOVA, followed by Dunnett's post-hoc test for multiple comparisons.

### Reporting summary

Further information on research design is available in the Nature Portfolio Reporting Summary linked to this article.

## Data availability

The meta-analysis in this study utilized publicly available sequence data from the NCBI SRA (accession numbers are provided in the Supplementary Information). The newly generated experimental data (transcriptome, amplicon, and whole-genome shotgun sequencing) have been deposited in the NCBI SRA under BioProject IDs: PRJNA1138414, PRJNA1138403, PRJNA1240447, PRJNA1240449, and PRJNA1138398. The raw metabolomics data have been deposited in the National Genomics Data Center (NGDC) OMIX repository under BioProject accession PRJCA052462 and are publicly available via: https://ngdc.cncb.ac.cn/omix/release/OMIX013393. Source data are provided with this paper.

## Code availability

The scripts for analysis are available on https://zenodo.org/records/16676212.

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

## Acknowledgements

We thank Jun Zhao (Nanjing Normal University) and Shaozhou Yang (Wenshan Sanqi Science and Technology Demonstration Park) for providing the rhizosphere soil of *Panax notoginseng*, as well as Xiang Li (Yangzhou University) for his help with DNA extraction from the *in planta* experiment samples. This work was supported by the National Natural Science Foundation of China (42207359 to L.S., 32270125 to H.-C.F., 32172661 to R.-F.Z. and 32270067 to Y.-L.Y.), Agricultural Science and

Technology Innovation Program to Y.-L.Y. and the China Postdoctoral Science Foundation (2021M693448 to L.S.).

## Author contributions

L.S. analyzed the metadata and wrote the main manuscript text. H.F. designed and conducted the greenhouse experiments and root exudate analyses. H.L., F.Y., X.Y., X.W. and P.L. performed the experimental studies. K.W., X.S., Y.L. and Q.S. provided critical suggestions and revisions to the manuscript. Y.Y. and R.Z. conceived, organized, and supervised the overall project.

## Competing interests

The authors declare no competing interests.
