## [Transparent Peer Review file · Nature Communications]

General variation in the Fusarium wilt rhizosphere microbiome

Corresponding Author: Professor Ruifu Zhang

Version 0:

Reviewer comments:

Reviewer #1

(Remarks to the Author)

The paper titled "Robust variation in the Fusarium wilt rhizosphere microbiome" by Su et al. explores how the microbial community in the rhizosphere of plants is affected by Fusarium wilt infection. The authors conducted a meta-analysis and comprehensive experiments to study the changes in bacterial composition and functional attributes in the rhizosphere of plants affected by Fusarium wilt. Their findings show that Fusarium infection significantly alters the bacterial community in the rhizosphere, with an enrichment of the genus *Flavobacterium*, which seems to have an antagonistic role against Fusarium pathogens. The study also highlights the enrichment of antioxidant functions related to sulfur metabolism and specific root exudates, such as tocopherol acetate, in the rhizospheres affected by the disease.

Overall, this is a well written and designed study. Strong points are the integration of a metanalysis with culture dependent approaches, which make the findings potentially reliable across different plant systems. The work has potential application in terms of managing soilborne pathogens. However, the limitations are a sole focus on *Flavobacterium* at the expense of other taxa, a lack of in-depth functional analysis, and insufficient handling of study heterogeneity. Strengthening these areas, along with broader ecological and mechanistic discussions, would enhance the impact of the study.

Specific comments:

L49-L51 – It would be useful to broadly introduce these mechanisms here e.g. root exudates, phytohormones, quorum sensing, induced systemic resistance (ISR), nutrient availability etc.

L82-86 - The concept of OAU is introduced to reduce the impact of multiple treatments on microbiome changes. However, the methods for addressing these confounding variables through the use of operational analysis units (OAUs) are not clearly justified. Whilst The use of OAUs to control for different treatments seems reasonable in principle, the section lacks details on how these units were defined and whether they were sufficient to control for the significant variation between studies.

L90 - While several studies were excluded due to low sequencing quality or insufficient metadata, the criteria for what constituted "sufficient" metadata are not explicitly defined.

L99-L101 - The study PRJNA562911 had over 70 samples and 8 OAUs, which the authors acknowledge could contribute to bias in the meta-analysis. However, the solution of randomly deleting three OAUs to reduce bias seems arbitrary and could potentially introduce randomness rather than systematically addressing the bias. The authors should consider alternative methods to address this bias, such as normalizing or weighting the data across studies, rather than random deletions.

L268-269 – For a study to be included in the meta-analysis, what do the authors mean by 'include a sequencing methodology'?

L274 – I assume the authors refer to 16S rRNA gene amplicon? Please specify

L280 – Do the authors mean that at position 100 (100bp) forward reads were truncated as the phred score fell below 30? Is this based on examining the demultiplexed quality plots of the sequences?

L280 – 284 - I am not familiar with this approach to denoise 16S rRNA reads to ASVs. There are elements in this pipeline which are standard but others I can't place. For example, quality filtering, dereplication and chimera removal seem standard but the issue I have is with `vsearch --usearch_global`, as this step is typically used for aligning sequences to a reference database, not for selecting ASVs. ASVs are exact sequence variants that are usually identified through denoising algorithms like those in DADA2 or Deblur. An alternative would be to cluster sequences based on an identity threshold into OTUs.

L296 - Salmon is typically used for transcript quantification. Ensure it's suitable for your metagenomic data.

L299 – I assume this is based on 16S, please specify. Is this based on minimum sequence depth after examining rarefaction plots or at a point where all samples have reached asymptote relative to richness?

L301 - Be cautious when using Chao1 as an estimation of richness for downstream differential abundance analysis. When samples approach the asymptote in a rarefaction plot, Chao1 becomes a reliable predictor of richness. If there are still exponential samples, like those with 5000 sequences, the estimate could vary significantly.

L307-308 –ADONIS can show significant differences between groups, but these differences might be due to differences in dispersion rather than differences in the centroid. Therefore, ADONIS should be followed with a multivariate homogeneity of group dispersions test. This helps to check if the assumption of homogeneity of variances is met. If groups have significantly different dispersions, it might affect the interpretation of the ADONIS results

Reviewer #2

(Remarks to the Author)

Title: Robust variation in the *Fusarium* wilt rhizosphere microbiome

General evaluation: This manuscript presented core microbial taxa in *Fusarium* wilt disease naturally suppressed soils. With various plant species, a bacterium, *Flavobacterium anhuiense* was selected as a responsible antagonistic agent against the wilt pathogen. Not only does the agent nominate, but the author also presents tocopherol acetate as an important root, which is to hire the agent when the plants are looking for a 'cry for help.'

Major concerns:

1. The authors did not generate the raw information. The data were obtained from NCBI and Google Scholar. Although the authors set up some criteria, the experimental and data analysis for the raw information may still not be even. The authors submitted this manuscript to [Nature Communications], and it is inappropriate for in silico data to be used for a high-ranking journal.
2. Again, the authors did not generate root exudate metabolite data.
3. How many *Flavobacterium* isolates were isolated with selective media, and how did you select the strain for genome seq. and further experiments? There is a lack of connection between the in silico and the strain K5.
4. Only 14 metabolites were evaluated to screen for *Flavobacterium* growth enhancement. In root exudate, there are numerous metabolites can exist and may relate to microbial growth.

Reviewer #3

(Remarks to the Author)

General comments:

The paper presents an interesting meta-analysis on the effects of *Fusarium* wilt on the plant microbiome. This study is useful as such interactions remain poorly understood and could lead to valuable applications in disease control.

In addition to the meta-analysis, the authors conducted further laboratory experiments to explore the mechanisms of plant-mediated fungal-bacterial interactions. These experiments provided biological validation for the meta-analysis, demonstrating that the strain *Fusobacterium anhuiense* K5 responds to *Fusarium* infection, as shown in various experiments. This aspect of the study is novel and well-executed.

However, I believe the paper could be improved linguistically and structurally. The language requires refinement, and there are limitations in how the methods and results are presented. Overall I would suggest that the authors are more detailed and careful when reporting their analytical approaches (for example, I assume vegan was used for multivariate statistics but no citation is to be found). Specifically, the authors should offer more detailed explanations regarding the meta-analysis. For instance, what criteria were used to group studies into Operational Analytical Units (OAUs)?

Additionally, throughout the paper, plant-pathogen-bacteria interactions are described solely as potential positive feedbacks for host health. However, it is equally possible that pathogenic or opportunistic bacteria could thrive during fungal infections, exacerbating the disease. This alternative outcome should at least be acknowledged in the introduction and discussion sections.

In addition, I am extremely hesitant to accept the term 'robust variation'. It might lead to confusion.

Specific comments:

Abstract:

Line 30. What of the strain was stimulated? The growth? It would be appropriate to clarify it.

Introduction:

Generally, the introduction covers the background and motivation for conducting this study. Nevertheless, in my opinion, the syntax should be improved, as many sentences are imprecise or misleading, furthermore, the text seem to lack a logical flow.

To give some examples Line 40. "Beneficial bacteria can be enriched in the rhizosphere of plants infected with pathogens". I think it is reasonable to first introduce the concept of dysbiosis (e.g. what described from line to 48) and then describe that among a dysbiosis certain changes could be beneficial for the host, while other could also be negative.

Line 49. "The microbial recruitment mechanism is the key theme in microbial ecology". This sentence sounds odd and misleading, I am sure microbial recruitment mechanisms are a very important topic of research of microbial ecology, but not the only key topic.

Line 50. Briefly explain why these tools can help understand recruitment mechanisms

Results:

Line 85. The authors used operational analysis units (OAU), which is a valid approach, but, in my opinion, the grouping criteria should be specified either here or in the methods section, as it is not clear how these groups were formed.

Line 87 to 93: The authors found a total of 37 studies where *Fusarium* was analyzed, 7 studies about Fungi, 6 having low quality, 9 lacking metadata and 2 adding bacterial wilt, were also removed. Now, $37 - (7 + 6 + 9 + 2) = 13$; but in the text it says that you only retained 9 studies. Am I missing something, or there is an error in the text?

Line 89. Explain the reasons to remove the fungal amplicon studies.

Line 109. What about observed richness? It is the fundamental index and the others rely on it

L117: I assume these statistical results correspond to PERMANOVA right?

Line 114. It appears to me that there are more appropriate statistical techniques to evaluate the effect of various predictors on community composition, such as the use of PERMANOVA with a strata argument. Why did the authors prefer to the PCoA axis and mixed effect models? If previously the effects of OAU were assessed with PERMANOVA, why not for infection status? That would allow a direct comparison of the explanatory power (R²)

Line 126. Is the random forest approach including OAU. If it is not I would reconsider the approach. OAU explains 76% of the variability, not including it in further analyses is a critical flaw

Line 136. What do you mean by "antioxidant catalog"? Is there a gene catalog for antioxidant genes you used? Or did you choose a specific KEGG group? Can you provide accession numbers in case?

Line 135 to 148. Please provide references that link genes to functions. Moreover, in the methods you say that you grouped genes into orthologs, but here you use the term genes. Are you talking about KEGG orthologs or genes? Please clarify and in case you are talking about orthologs, add the KO accession number.

Discussion:

Line 200. I am somehow not understanding the logical link between how the listed gene categories changing contribute to positive feedback for the host health. I understand degradation of chitin, but why would you expect lower degradation on fumarate and malate in infected communities? Could you clarify in the text why you expect these functions to be relevant in host/microbiome interactions?

Line 205. "to achieve this goal, we performed...", you already said in the discussion that you performed a metanalysis. No need to repeat it once again, I think at this point of the paper it is clear to the reader.

Line 218. I do not understand why you speak about functions both between lines 193 to 202, then you start speaking about *Flavobacterium*, and then you go back speaking about functions again; as for the introduction, I would advise to restructure the text.

Line 218. What do you mean by "generalizing the microbiome functions"?

Methods:

I suggest adding a description of how OAU were determined to clarify how data was grouped together.

Line 259. I think a list of the studies, including if they were used or not, should be reported in supplementary information. I see that similar information is already provided in supplementary table S1 and S2; nevertheless, I think it is still difficult to understand which studies were originally filtered and which were kept, and the corresponding publication.

Line 262 to 265. Put the complete search terms in quotation marks

Line 295: It is unclear to me what "which were then predicted for overlapping groups with Prokka". Prokka is an annotation tool, what does it mean, it predicts overlapping groups?

Can it be that you mean "open reading frames", or "gene coding sequences"?

Line 301: "Statistical analysis was performed via the Wilcoxon test" what statistical analysis? To me it seems you used either linear mixed effect models or random forest. What other tests did you run?

Line 303. Imer is not a function of the package lmerTest

Line 307. adonis is the vegan function, not the method

Line 310. I am not an expert in reporting pot experiments, but it seems to me that the description is lacking details important for reproducibility. For example, how old were the seedlings? Were they grown with the pathogen for only 10 days?

Line 318. How many ml for how much soil?

Line 324. Was the genome sequenced by you? There are no details regarding the sequencing. How was it conducted? . On which platform? I suppose you used short reads by the assembler you used, but I should not guess, it should be specified.

Line 343: Please report which studies were reviewed.

Line 358. How were these root exudates made? In vitro, or purchased? Is the description on the original paper you quote?

Line 375. How was the sequencing made? Which platform?

Line 428. Which data is normally distributed?

Figures:

Figure 1. Improve description, what does the top right p-value indicate? Supplementary Fig. 2 should be part of Fig. 1 to show the reader the large effect of OAU. Any stats to support the statement of 'clearly distinct clusters'?

Figure 2A. Please correct x-axis label. Also what relative abundance is represented? Mean across all studies? Differences between healthy and infected? Please clarify, it is unclear.

Figure 3 is really hard to interpret. I would suggest transforming it into a table with full name, KEGG category and accession numbers, direction, intensity and statistical significance of the change (or some other visualization approach).

Reviewer #4

(Remarks to the Author)

Version 1:

Reviewer comments:

Reviewer #1

(Remarks to the Author)

The author's response is sufficient, and I have no further comments to add.

(Remarks on code availability)

Reviewer #2

(Remarks to the Author)

Dear Editor

The authors showed many effects in the revised manuscript, such as adding new metabolite experiments. However, there is no in planta assay to validate the findings. The authors only suggested a proposal concept in Fig. S6. Nature Communications is a high-ranking journal, and I believe the keystone taxa's result in planta assay is necessary.

(Remarks on code availability)

Reviewer #5

(Remarks to the Author)

This is a revised version of a manuscript previously submitted to Nature Communications. I was not involved in the first round of review, so my assessment focuses primarily on whether the authors have adequately addressed the reviewers' previous comments.

Overall, I find that the manuscript has been significantly improved in this revised version.

However, my main concern is related to the availability of the code. The authors have provided R scripts via a Zenodo repository, but they have not included the input data or sufficient documentation to explain how to run the scripts. These are essential components for ensuring reproducibility and should be addressed before publication.

(Remarks on code availability)

As stated in the "Comments for Author" section, the authors should provide the input data and appropriate documentation to ensure the reproducibility of the code.

Reviewer #6

(Remarks to the Author)

The paper "General variation in the *Fusarium* wilt rhizosphere microbiome" by Su et al presents a meta-analysis and experimental validation on how *Fusarium* wilt infection affects the rhizosphere bacterial community and root exudates. It focuses particularly on:

Changes in bacterial community composition and function

Role of root exudates (especially tocopherol acetate)

Isolation and behavior of *Flavobacterium* strains, which are identified as the culprit in *Fusarium* wilt rhizosphere microbiomes.

Identifying *Flavobacterium* and in particular *F. anhuiense* as significant component of *Fusarium* wilt and Tocopherol acetate as a growth stimulant are the central findings. These are novel findings and not trivial, as tocopherol isn't known to be a common microbial growth stimulant.

Major revisions:

The observed diversity increase in disease samples seems counterintuitive, as dysbiosis commonly is characterized by diversity decrease. As such it would be great if the authors could elaborate on this counterintuitive circumstance, at the very least in the Discussion section.

For presentational purposes, it would be desirable to provide visualizations that allow the inspection of the microbial community composition on various taxonomic ranks. QIIME2 for example provides web based tools for this matter (stacked bar diagrams). It allows the sorting of samples based on particular genera or “species” of interest (in this case: the top predictors for diseased classes like *Flavobacterium*, *Variovorax*, *Stenotrophomonas* etc.). Likewise, I would have expected a standard visualization of all alpha diversity scores, e.g. as box plots.

None of the used alpha diversity measures take phylogenetic distance into account. It would be desirable to include at least one, like Faith’s phylogenetic distance.

Finally it is very common and useful to support microbiome analysis with beta-diversity analysis. The authors perform PCoA, which is a rather technical term and beta-diversity should be used where possible. The details of beta-diversity are not well-described. The code for Figure 1 seems to use Bray-Curtis, but this is not described in the manuscript/Methods section.

UniFrac[1] is recommendable as it takes phylogenetic distances of community constituents into account. The provided code for Figure 1 seems to use the Bray-Curtis distance. Various visualization methods like a clustered heatmap with hierarchical clustering dendrograms can give a formidable overview over all samples and how they relate to each other.

A particular issue is that the partitioning of the data into 25 “operational analysis units” (OAUs), many of them containing below 10 data points (some have only 6), reduces statistical power. 6 samples in a PCoA cluster easily by chance. It is also not clear according to which criteria the OAUs are defined. The supplementary table(s)

Line 154: the text doesn’t say what exactly the classification task is, i.e. what are the features, how many samples were used (was the learning done for each OAU separately?) and what are the exact classes. The latter are likely to be healthy vs diseased and the former the genera present in the respective sample but it’s best to state that explicitly and exactly in the Methods section.

The choice of method to identify the best predictors is not well motivated. Random Forests are common but far from the only method to identify predictors. First of all, it should be demonstrated that the formulated Machine learning task is actually doing a reasonable job, i.e. accuracy values should be reported. Generally, SHAP/TreeSHAP[2] is currently the best practice to determine feature importance, as it associates high explainability/interpretability with additive (and other desirable mathematical) characteristics to features.

The reproducibility of the code is severely hampered:

The version numbers for the required packages (most importantly: *vegan*) are not provided

The code doesn’t run out of the box, as it assumes the presence of certain objects (*pl* in line 16, *eam_gr* in line 17 in Fig.1.R, *marker_ok_u* in) that are not instantiated. I would assume that *eam_gr* contains some preprocessed microbial abundance data, but that is not in the provided repository. The code would benefit from better commenting and more intuitive variable names

Non-standard libraries like *lmerTest* don’t install via the `install.packages` command. It is therefore important to provide installation instructions

Given point 1-3. above, it is recommendable to provide a better environment for code reproducibility, like a containerized version including all needed code, libraries and ideally also data (or links/download instructions to (SRA) data repositories

Supplementary Material:

Table S5: the table contains a column “References”, but doesn’t actually provide complete references (no journal, author etc). The references are often also not referring to seminal work, e.g.: Role of *Williamsia* and *Segniliparus* in human infections with the approach taxonomy, cultivation, and identification methods is only cited 7 times.

The “Functions” column needs to be revised. It is out of my expertise, but a quick check on *Williamsia* did not confirm it to be a well known “People pathogen”, possibly it can be described as an opportunistic pathogen for humans.

It would be very useful if the OAU tables also contained the number of samples and the number of different studies these samples are coming from.

Minor revisions:

Line 590 most likely contains a typo. GIMI should be Gini.

The figures at the end (after line 845, page 42) are not numbered nor captioned.

Table S4 Significant test of principal coordinate analysis in each OAU.

Should be: Table S4 Significance test ...

References

[1] Lozupone, Catherine, Micah Hamady, and Rob Knight. “UniFrac – An Online Tool for Comparing Microbial Community Diversity in a Phylogenetic Context.” *BMC Bioinformatics* 7, no. 1 (August 7, 2006): 371. <https://doi.org/10.1186/1471-2105-7-371>.

[2] Lundberg, Scott, and Su-In Lee. "A Unified Approach to Interpreting Model Predictions." arXiv, November 25, 2017. <https://doi.org/10.48550/arXiv.1705.07874>.

(Remarks on code availability)

The reproducibility of the code is severely hampered:

The version numbers for the required packages (most importantly: vegan) are not provided

The code doesn't run out of the box, as it assumes the presence of certain objects (pl in line 16, eam_gr in line 17 in Fig.1.R, marker_ok_u in Fig.2.R) that are not instantiated. I would assume that eam_gr contains some preprocessed microbial abundance data, but that is not in the provided repository. The code would benefit from better commenting and more intuitive variable names

Non-standard libraries like lmerTest don't install via the install.packages command. It is therefore important to provide installation instructions

Given point 1-3. above, it is recommendable to provide a better environment for code reproducibility, like a containerized version including all needed code, libraries and ideally also data (or links/download instructions to (SRA) data repositories

Version 2:

Reviewer comments:

Reviewer #2

(Remarks to the Author)

Thank you for your revised submission, which incorporates the reviewers' suggestions. I would like to provide the following comments regarding the newly added in planta assay results.

In Figure 6, the pathogen density appears to be statistically reduced by treatment with strain K5. However, I find three critical issues:

1. The pathogen density remains above 10^6 copy numbers, which is still sufficient to cause disease. Moreover, the copy number presented by the authors is not pathologically meaningful. Results based on copy number should be converted into cfu per gram. It is well known that Fusarium can occur disease development when present at 10^5 cfu per gram.
2. Pathogen density was measured only at 24 and 72 hours, which does not reflect the ecological behavior of Fusarium. Typically, Fusarium requires more than 7 days to colonize and penetrate plant roots. Therefore, pathogen density should be measured at additional time points over an extended period.
3. More important, the actual disease incidence and suppression efficacy remain unaddressed in the manuscript.

(Remarks on code availability)

Reviewer #5

(Remarks to the Author)

The authors have responded positively to my previous comments and have revised the manuscript accordingly.

(Remarks on code availability)

The authors have provided a Zenodo repository containing the code necessary to reproduce the results.

Reviewer #6

(Remarks to the Author)

I would like to thank the authors for meticulously addressing all my concerns. I have no further concerns.

(Remarks on code availability)

Previously I have tried to run the code and had uttered some concerns. Now all subdirectories are equipped with respective README files and much improved installation instructions

**Dear Reviewers,**

**Thank you for giving us the opportunity to revise this manuscript. We greatly**
**appreciate the time and effort you have invested in providing valuable feedback,**
**which has greatly enhanced this work.**

**Major modifications were listed as follows.**

**(1). Amplicon sequencing data of the rhizosphere bacterial community in**
***Fusarium*-infected tomato and *Panax notoginseng* were generated.**

**(2). Metabolite data for tomato root exudates under *Fusarium* invasion were**
**generated.**

**(3). The effects of 35 root exudate metabolites on the growth of *Flavobacterium***
***anhuiense* were determined**

**(4). The deleting OAU data were added, and the meta data were re-analyzed.**

**(5). The manuscript was carefully checked and professionally edited by AJE for**
**language quality.**

**Thanks again for your helpful suggestions!**

**Reviewers' comments:**

Reviewer #1 (Remarks to the Author):

The paper titled "Robust variation in the *Fusarium* wilt rhizosphere microbiome" by
Su et al. explores how the microbial community in the rhizosphere of plants is
affected by *Fusarium* wilt infection. The authors conducted a meta-analysis and
comprehensive experiments to study the changes in bacterial composition and
functional attributes in the rhizosphere of plants affected by *Fusarium* wilt. Their
findings show that *Fusarium* infection significantly alters the bacterial community in
the rhizosphere, with an enrichment of the genus *Flavobacterium*, which seems to
have an antagonistic role against *Fusarium* pathogens. The study also highlights the
enrichment of antioxidant functions related to sulfur metabolism and specific root
exudates, such as tocopherol acetate, in the rhizospheres affected by the disease.

(1) Overall, this is a well written and designed study. Strong points are the integration
of a metanalysis with culture dependent approaches, which make the findings
potentially reliable across different plant systems. The work has potential application
in terms of managing soilborne pathogens. However, the limitations are a sole focus
on *Flavobacterium* at the expense of other taxa. a lack of in-depth functional analysis,
and insufficient handling of study heterogeneity. Strengthening these areas, along
with broader ecological and mechanistic discussions, would enhance the impact of the
study.

**Answer: Thank you for your thoughtful suggestions and positive feedback on**
**our study. We appreciate your recognition of the integration of meta-analysis**
**with culture-dependent approaches, as well as the potential applications of our**
**findings in managing soilborne pathogens.**

**In the revised version, we have addressed the limitations you highlighted by**
**taking the following steps:**

(1) **We determined the compositions and functions of *Flavobacterium* and other**
**biomarkers that showed significant changes in both the meta-data and our**
**validation data (amplicon sequencing data generated in our study related to**
***Fusarium*-infected tomato and *Panax notoginseng* rhizosphere bacterial**
**communities).**

(a) **Biomarkers were selected as stable significant predictors based on**
**random forest analysis (significant in both MeanDecreaseGini and**
**MeanDecreaseAccuracy indices). In addition, they should significantly enrich in**
**the diseased rhizosphere as tested by a linear mixed-effects model. The genera**
***Variovorax*, *Flavobacterium*, *Stenotrophomonas*, *Rhizobium*, *Williamsia*,**
***Aeromonas*, *Novosphingobium*, and *Micromonospora* met this criterion.**

(b) **The functions of these genera were identified from published literature,**

revealing that *Flavobacterium*, *Stenotrophomonas*, and *Micromonospora* can
inhibit *Fusarium* growth.

(c) Only *Flavobacterium* was significantly enriched in both the meta-data and
our amplicon sequencing data, making it the focus of further analysis.

**(2) Strengthening Study Heterogeneity:**

To address study heterogeneity, we added the deleting OAU data and re-
analyzed the meta-data. Additionally, we generated the amplicon sequencing
data to further support the meta-analysis results.

**(3) Broader Ecological and Mechanistic Discussions:**

We have expanded the discussion to include broader ecological implications and
mechanistic insights, with a particular focus on the role of *Flavobacterium* in
suppressing *Fusarium* and its underlying recruitment mechanisms

(2) Specific comments: L49-L51 – It would be useful to broadly introduce these
mechanisms here e.g. root exudates, phytohormones, quorum sensing, induced
systemic resistance (ISR), nutrient availability etc.

**Answer: Thank you for your suggestion to broadly introduce the mechanisms**
**related to microbial recruitment in the rhizosphere. In the introduction, we have**
**included a discussion of root exudates, phytohormones, quorum sensing, induced**
**systemic resistance (ISR), and nutrient availability. Specifically, to align with the**
**focus of our study, we have emphasized the recruitment mechanisms associated**
**with changes in root exudates induced by pathogen invasion.**

Elucidating microbial recruitment mechanism helps to clarify the causes of
changes in diseased rhizosphere microbial composition, which is an important topic in
microbial ecology. Root exudates ⁹, phytohormones ¹⁰, quorum sensing ¹¹, induced
systemic resistance ¹², and nutrient availability ¹³ significantly influence the
recruitment of rhizosphere bacterial communities. Among these, root exudates serve
as a critical bridge linking plants and rhizosphere microorganisms. Most of the
pathways discussed above involve the influence of root exudates on microbe
recruitment. During this process, microbes are initially attracted by specific
components of root exudates and colonize the root surface through chemotaxis.
Subsequently, they utilize root exudates for rapid proliferation and establish stable
colonization on the rhizoplane by forming biofilms ¹⁴. Moreover, pathogen invasion
could significantly alter the composition of root exudates. For example, the
metabolites such as tocopherol acetate, citrulline, galactitol, octadecylglycerol, and
behenic acid were greatly enriched in the rhizosphere of plants infected by *Fusarium*
¹⁵. *Fusarium* infection also triggered phenolic acids and organic acids of succinic acid
and oxalic acid production in the watermelon root exudates ¹⁶. These enriched
metabolites may contribute to the rhizosphere recruitment of beneficial microbes,

which protect plants from pathogen invasion. For instance, *Arabidopsis thaliana*
infected by *P. syringae* pv tomato induces the secretion of malic acid that stimulates
biofilm formation of *Bacillus subtilis*¹⁷. Furthermore, pathogen invasion can alter the
functional profile of the rhizosphere microbiome, reflecting changes in functional
genes related to nutrient metabolism. For example, the relative abundances of genes
associated with amino acid transport and metabolism decreased in the rhizosphere of
*Fusarium-wilt P. notoginseng* suggesting a depletion of microbes capable of utilizing
these metabolites¹⁸. Overall, the combined application of metagenomics and
metabolomics provides a systematic approach to unravel the recruitment mechanisms
of rhizosphere microbes, offering deeper insights into plant-microbe interactions
under pathogen stress.

(3) L82-86 - The concept of OAU is introduced to reduce the impact of multiple
treatments on microbiome changes. However, the methods for addressing these
confounding variables through the use of operational analysis units (OAUs) are not
clearly justified. Whilst The use of OAUs to control for different treatments seems
reasonable in principle, the section lacks details on how these units were defined and
whether they were sufficient to control for the significant variation between studies.

**Answer: We appreciate the opportunity to clarify the concept of OAUs.**

**When performing combined analyses across different studies, random effects**
**such as sampling locations, sampling times, and other confounding factors can**
**obscure the differences between control and treatment groups.**

**To address this, we employed the Generalized Linear Mixed Model (GLMM), a**
**widely used statistical model in meta-analyses that accounts for both fixed effects**
**(e.g., diseased vs. healthy plants) and random effects (e.g., sampling locations,**
**sampling times). For instance, Bisanz et al. (2019) used GLMM to analyze the**
**effects of high- and low-fat diets on the gut microbiome across studies, treating**
**diet as a fixed effect and individual studies as random effects.**

**However, soil is an open environment where the microbiome is highly sensitive to**
**external factors. Within a single study, variations in sampling locations and**
**times can significantly disrupt the comparison between healthy and diseased**
**rhizosphere microbiomes.**

**To address this, we introduced the concept of Operational Analysis Units**
**(OAUs), which refine the analysis by subdividing samples into subgroups until**
**comparisons between diseased and healthy rhizosphere microbiomes can be**
**made under the same conditions (e.g., same sampling location and time within a**
**study). Samples from the same sampling location or time are grouped into one**
**OAU. These subgroups are defined as OAUs (see Fig. S6). This approach**
**provides a more intuitive way to compare diseased and healthy microbiomes**

**under controlled conditions.**

**To demonstrate the importance of OAUs, we compared two scenarios:**

**Using OAUs as random effects: Pathogen infection significantly affected**
**bacterial composition ($P < 0.001$).**

**Without using random effects: Pathogen infection did not significantly affect**
**bacterial composition ($P = 1$).**

**Importantly, we have not introduced any new statistical models. Instead, we**
**refined the random effects to enable comparisons between control and treatment**
**groups under consistent conditions within a study. The OAU method aligns with**
**the principles of GLMM while providing a more intuitive framework for**
**analyzing microbiome changes.**

**We hope this explanation clarifies the rationale and methodology behind the use**
**of OAUs in our study. Thank you again for your valuable feedback.**

(4) While several studies were excluded due to low sequencing quality or insufficient
metadata, the criteria for what constituted "sufficient" metadata are not explicitly
defined.

**Answer: Corrected.**

**Insufficient metadata was defined as follows:**

**The metadata could not clearly distinguish between healthy and diseased plants.**

**Studies lacking essential information, such as sampling conditions, treatment**
**details were also excluded. These excluded studies have been listed in Table S1.**

(5) L99-L101 - The study PRJNA562911 had over 70 samples and 8 OAUs, which
the authors acknowledge could contribute to bias in the meta-analysis. However, the
solution of randomly deleting the OAUs to reduce bias seems arbitrary and could
potentially introduce randomness rather than systematically addressing the bias. The
authors should consider alternative methods to address this bias, such as normalizing
or weighting the data across studies, rather than random deletions.

**Answer: Corrected.**

**In the revised analysis, we have re-included the previously deleted OAUs and**
**normalized the data to ensure a more balanced representation across studies.**
**The re-analyzed results remain consistent with our previous findings, confirming**
**the robustness of our conclusions.**

(6) L268-269 – For a study to be included in the meta-analysis, what do the authors

mean by 'include a sequencing methodology'?

**Answer: By "include a sequencing methodology," we mean that the published**
**studies must provide amplicon sequencing data and describe the sequencing**
**analysis methods used to investigate the effects of *Fusarium* infection on the**
**rhizosphere microbiome. This ensures that the studies included in our meta-**
**analysis provide reliable data for comparative analysis.**

**For clarity, we have rephrased the text as follows:**

"For a study to be included in the meta-analysis, it had to include amplicon
sequencing data and a detailed sequencing analysis method for studying the effects of
*Fusarium* infection on the rhizosphere microbiome."

(7) L274 – I assume the authors refer to 16S rRNA gene amplicon? Please specify

**Answer: Thank you for pointing this out. Yes, we refer to 16S rRNA gene**
**amplicon sequencing data, which were downloaded from the Sequence Read**
**Archive (SRA). We have revised the text to specify this:**

"Amplicon sequencing data (16S rRNA gene) were downloaded from the SRA."

(8) L280 – Do the authors mean that at position 100 (100bp) forward reads were
truncated as the phred score fell below 30? Is this based on examining the
demultiplexed quality plots of the sequences?

**Answer: Thank you for your question regarding the sequence processing steps.**
**We appreciate the opportunity to clarify this point.**

**The truncation of forward reads at 100 bp was not directly associated with the**
**Phred score (<30). Instead, the processing involved two distinct steps:**

**We first used the Trim Galore tool to remove base sequences with a Phred**
**quality score below 30.**

**Subsequently, based on the understanding that 90-bp sequences are sufficiently**
**long to reveal detailed patterns of community structure, we retained only the**
**forward reads and truncated them to 100 bp for consistency and efficiency in**
**downstream analysis.**

(9) L280 – 284 - I am not familiar with this approach to denoise 16S rRNA reads to
ASVs. There are elements in this pipeline which are standard but others I can't place.
For example, quality filtering, dereplication and chimera removal seem standard but
the issue I have is with vsearch --usearch_global, as this step is typically used for
aligning sequences to a reference database, not for selecting ASVs. ASVs are exact

sequence variants that are usually identified through denoising algorithms like those
in DADA2 or Deblur. An alternative would be to cluster sequences based on an
identity threshold into OTUs.

**Answer: Thank you for your detailed feedback regarding the denoising**
**approach for 16S rRNA reads. We appreciate your suggestions and have**
**clarified the methodology in the revised manuscript.**

**To address your concerns:**

**We used vsearch --cluster_unoise to obtain Amplicon Sequence Variants (ASVs),**
**not vsearch --usearch_global. This step has been corrected in the manuscript.**

**The vsearch tool is widely used for analyzing 16S rRNA gene amplicon**
**sequencing data and is effective for identifying ASVs.**

**The abundances of ASVs were determined using vsearch --usearch_global,**
**which aligns sequences to the ASV reference set to quantify their abundances.**

**The revised text now reads:**

"Finally, ASVs were selected via vsearch --cluster_unoise and annotated against the
RDP database at a cutoff of 0.8. The abundances of ASVs were obtained using
vsearch --usearch_global."

**Thank you again for your valuable feedback, which has improved the accuracy**
**and clarity of our methodology.**

**(10) L296 - Salmon is typically used for transcript quantification. Ensure it's suitable**
**for your metagenomic data.**

**Answer: We have carefully reviewed this step and confirmed its suitability.**

**Salmon has been integrated into the EasyMetagenome pipeline, a user-friendly**
**and flexible tool designed for shotgun metagenomic analysis in microbiome**
**research. Within this pipeline, Salmon is used for quantifying gene abundances,**
**leveraging its efficiency and accuracy in handling large-scale metagenomic**
**datasets.**

**For further details, please refer to the following reference:**

**EasyMetagenome: A user-friendly and flexible pipeline for shotgun metagenomic**
**analysis in microbiome research.**

**We hope this clarification addresses your concern. Thank you again for your**
**valuable feedback.**

(11) L299

(a) The samples were rarefied to 5000 sequences to generate alpha diversity. I assume
this is based on 16S, please specify.

**Answer: Yes, it is based on 16S rRNA gene amplicon sequencing data.**

(b) Is this based on minimum sequence depth after examining rarefaction plots or at a
point where all samples have reached asymptote relative to richness?

**Answer: Thank you for your question regarding the rarefaction process.**

**The rarefaction to 5,000 sequences was based on the following steps:**

**We removed samples with a sequence depth of less than 5,000, as this threshold**
**is commonly used in published meta-analyses to ensure sufficient sequencing**
**depth for reliable analysis.**

**For the remaining samples, we used the minimum sequence depth among them**
**to perform rarefaction, ensuring consistency and comparability across all**
**samples.**

**We have revised the manuscript to specify this process:**

"The amplicon samples with a sequence depth of less than 5,000 were removed. The
remaining samples were rarefied to the minimum sequence depth to ensure consistent
analysis."

(12) L301 - Be cautious when using Chao1 as an estimation of richness for
downstream differential abundance analysis. When samples approach the asymptote
in a rarefaction plot, Chao1 becomes a reliable predictor of richness. If there are still
exponential samples, like those with 5000 sequences, the estimate could vary
significantly.

**Answer: Thank you for your suggestion regarding the use of the Chao1 index for**
**estimating richness. We acknowledge that Chao1 may not be reliable when**
**samples have not reached the asymptote in rarefaction plots, particularly for**
**exponential samples with limited sequencing depth (e.g., 5,000 sequences).**

**We found certain exponential samples in our meta-data, which could lead to**
**significant variability in Chao1 estimates. To ensure robust results, we have**
**removed the Chao1 index results from our analysis.**

(13) L307-308 –ADONIS can show significant differences between groups, but these
differences might be due to differences in dispersion rather than differences in the
centroid. Therefore, ADONIS should be followed with a multivariate homogeneity of
group dispersions test. This helps to check if the assumption of homogeneity of
variances is met. If groups have significantly different dispersions, it might affect the
interpretation of the ADONIS results

**Answer: We agree that differences identified by ADONIS could be due to**
**variations in dispersion rather than centroid differences. To address this, we**
**performed a multivariate homogeneity of group dispersions test using the**
**betadisper function in R.**

**The result of the betadisper test was significant ($P = 0.001$), indicating that the**
**observed differences might indeed be influenced by variations in dispersion. To**
**ensure robust and reliable results, we have removed these results from the**
**manuscript.**

**Thank you again for your valuable feedback, which has improved the rigor of**
**our analysis.**

Reviewer #2 (Remarks to the Author):

**Title: Robust variation in the Fusarium wilt rhizosphere microbiome General**
**evaluation: This manuscript presented core microbial taxa in Fusarium wilt disease**
**naturally suppressed soils. With various plant species, a bacterium, *Flavobacterium***
***anhuiense* was selected as a responsible antagonistic agent against the wilt pathogen.**
**Not only does the agent nominate, but the author also presents tocopherol acetate as**
**an important root, which is to hire the agent when the plants are looking for a ‘cry for**
**help.’**

**Major concerns:**

**(1) The authors did not generate the raw information. The data were obtained from**
**NCBI and Google Scholar. Although the authors set up some criteria, the**
**experimental and data analysis for the raw information may still not be even. The**
**authors submitted this manuscript to [Nature Communications], and it is inappropriate**
**for in silico data to be used for a high-ranking journal.**

**Answer: Thank you for your suggestions regarding the use of in silico data. We**
**appreciate your feedback.**

**We acknowledge the limitations of relying solely on publicly available data. To**
**strengthen our findings, we have taken steps to address this concern.**

**(a) In the revised version of the manuscript, we have generated new**
**experimental data, including: [1] Amplicon sequencing data of the rhizosphere**
**bacterial community in tomato and *Panax notoginseng*. [2] Root exudate**
**metabolite data from diseased and healthy tomato plants.**

**(b) Our experimental data aligns with the results from the meta-analysis.**
**Specifically: [1] *Fusarium* infection significantly altered the bacterial community**
**in both the meta-data and our data. [2] Certain genus-level biomarkers**
**identified in the meta-analysis were also significantly enriched in our data.**
**Notably, *Flavobacterium* was significantly enriched in both the meta-data and**
**our data.**

**By incorporating these new experimental findings, we have strengthened the**
**robustness and reliability of our study. Thank you again for your valuable**
**feedback, which has significantly improved the quality of our manuscript.**

**(2) Again, the authors did not generate root exudate metabolite data.**

**Answer: Thank you for your comment regarding the root exudate metabolite**
**data. We have generated experimental data for root exudates in the new version.**

**In our experiment, we collected root exudates from both healthy and *Fusarium*-**

infected tomato plants. The results demonstrated that *Fusarium* invasion
significantly altered the composition of root exudates. Specifically, tocopheryl
acetate, a dominant component, was significantly enriched in *Fusarium*-infected
treatments. This finding aligns with and supports the results from our meta-
analysis.

Thank you again for your valuable feedback, which has helped improve the
clarity and rigor of our manuscript.

(3) How many *Flavobacterium* isolates were isolated with selective media, and how
did you select the strain for genome seq. and further experiments? There is a lack of
connection between the in silico and the strain K5.

Answer: Thank you for your question regarding the isolation and selection of
*Flavobacterium* strains. We appreciate the opportunity to clarify our
methodology.

The selective media PSR2A-C/T was not specific to *Flavobacterium*. As shown in
previous studies, other microbes such as *Chryseobacterium* were also present in
the media. We isolated 36 yellow colonies, but only two of these were identified as
*Flavobacterium* (*Flavobacterium anhuiense* and *Flavobacterium daejeonense*).

Selection of *Flavobacterium anhuiense* K5:

To align with our study, we mapped the 16S rRNA genes of the two
*Flavobacterium* isolates to the ASVs from our tomato rhizosphere amplicon data.
We found that only *Flavobacterium anhuiense* was dominant and significantly
enriched in the diseased rhizosphere. Therefore, we selected *Flavobacterium*
*anhuiense* K5 for genome sequencing and further experiments.

This approach ensures a clear connection between the in silico analysis and the
experimental validation using strain K5. We hope this explanation addresses
your concern. Thank you again for your valuable feedback, which has improved
the clarity of our manuscript.

(4) Only 14 metabolites were evaluated to screen for *Flavobacterium* growth
enhancement. In root exudate, there are numerous metabolites can exist and may
relate to microbial growth.

Answer: Thank you for your valuable suggestion. We fully acknowledge your
point that root exudates contain numerous metabolites, many of which may
influence microbial growth. We agree that evaluating only a subset of
metabolites could limit the comprehensiveness of our study.

**In response to your comment, we have expanded our analysis to include a**
**broader range of metabolites. Guided by the "Cry for Help" theory, we focused**
**on metabolites significantly enriched under *Fusarium* infection. Specifically, we**
**extended our scope from metabolites co-enriched in at least two studies (14**
**components) to those enriched in any study (35 components).**

**Our results revealed that numerous metabolites significantly stimulated the**
**growth of *Flavobacterium anhuiense* K5 at 10 h, 19 h, and 30 h. Among these,**
**tocopheryl acetate consistently showed the highest stimulation level across all**
**time points.**

**Thank you again for your insightful feedback, which has greatly improved the**
**quality of our work.**

Reviewer #3 (Remarks to the Author):

General comments:

The paper presents an interesting meta-analysis on the effects of Fusarium wilt on the
plant microbiome. This study is useful as such interactions remain poorly understood
and could lead to valuable applications in disease control.
In addition to the meta-analysis, the authors conducted further laboratory experiments
to explore the mechanisms of plant-mediated fungal-bacterial interactions. These
experiments provided biological validation for the meta-analysis, demonstrating that
the strain Fusobacterium anhuiense K5 responds to Fusarium infection, as shown in
various experiments. This aspect of the study is novel and well-executed.

(1) However, I believe the paper could be improved linguistically and structurally.
The language requires refinement, and there are limitations in how the methods and
results are presented.

**Answer:** Thank you for your thoughtful and constructive feedback on our
manuscript. We greatly appreciate your recognition of the novelty and
significance of our study, particularly the laboratory experiments that provide
biological validation for the meta-analysis.

**In response to your comments, we have taken the following steps to improve the
manuscript:**

**Language Refinement:** We have carefully revised the language throughout the paper
to enhance clarity and readability. Additionally, the manuscript has been
professionally edited by AJE (American Journal Experts) to ensure high-quality
English.

**Structural Adjustments:** We have reorganized the manuscript structure to improve
flow and coherence, particularly in the methods and results sections, which have been
enriched based on your suggestions.

**We believe these revisions have strengthened the manuscript. Thank you again
for your valuable insights and support.**

(2) Overall I would suggest that the authors are more detailed and careful when
reporting their analytical approaches (for example, I assume vegan was used for
multivariate statistics but no citation is to be found).

**Thank you for your additional comment regarding the reporting of our
analytical approaches. We appreciate your attention to detail, which has helped
501 us improve the manuscript further.**

**In response to your suggestion, we have carefully reviewed and revised the**
**description of our analytical methods. Specifically:**

- **1. We have provided more detailed and accurate reporting of the analytical**
**approaches used in the study, including the use of the *vegan* package for**
**multivariate statistics, which has now been properly cited.**
- **2. We have ensured that all other analytical tools and methods are clearly**
**and comprehensively described to enhance transparency and**
**reproducibility.**

(3) Specifically, the authors should offer more detailed explanations regarding the
meta-analysis. For instance, what criteria were used to group studies into Operational
Analytical Units (OAUs)?

**Answer: Corrected.**

**We have addressed the comment regarding the meta-analysis by providing a**
**more detailed explanation of the criteria used to group studies into Operational**
**Analytical Units (OAUs).**

**The updated explanation is as follows:**

"Within a single study from our meta-dataset, variations in sampling locations and
526 times can significantly disrupt the comparison between healthy and diseased
rhizosphere microbiomes. To address this, we introduced the concept of Operational
Analysis Units (OAUs), which refine the analysis by subdividing samples into
subgroups until comparisons between diseased and healthy rhizosphere microbiomes
can be made under the same conditions (e.g., same sampling location and time within
a study). Samples from the same sampling location or time are grouped into one OAU.
These subgroups are defined as OAUs."

**For more detailed information about the criteria used to form OAUs, please**
**refer to our response to Reviewer 1, Question 3, line 127.**

**We appreciate your thorough review and hope that these clarifications meet**
**your expectations.**

(4) Additionally, throughout the paper, plant-pathogen-bacteria interactions are
described solely as potential positive feedbacks for host health. However, it is equally
possible that pathogenic or opportunistic bacteria could thrive during fungal infections,
exacerbating the disease. This alternative outcome should at least be acknowledged in
the introduction and discussion sections.

**Answer:**

**We appreciate your insightful comments, which have helped us improve the**
**quality of our paper.**

**Regarding point (4), we agree that it is important to acknowledge the potential**
**for pathogenic or opportunistic bacteria to thrive during fungal infections, which**
**could exacerbate disease. As suggested, we have added this consideration to the**
**introduction section. The revised text now reads:**

"The plant microbiome is referred to as the second genome of a plant and plays
important roles in host health ¹. Imbalance of the plant microbiome, known as
dysbiosis, significantly influences plant health ². The disruption of potential
antagonistic bacteria, such as Firmicutes and Actinobacteria, in the tomato
rhizosphere by vancomycin may lead to microbiome dysbiosis, potentially enriching
pathogens such as *Ralstonia solanacearum* and *Fusarium oxysporum* ³. Similarly,
invasion by the tomato soil-borne pathogen *R. solanacearum* is found in many
rhizosphere ecological niches, significantly decreasing the relative abundances of
antagonistic bacteria and the bacterial diversity that may contribute to other fungal
pathogen infections ⁴. Plants and microbes have coevolved for hundreds of millions of
565 years and formed a very sophisticated and beneficial system of communication.
Certain beneficial bacteria can be enriched in the rhizosphere of plants infected with
*Fusarium* pathogens, potentially contributing to disease suppression or plant health
improvement ^{5, 6}. *Bacillus* and *Sphingomonas* are significantly enriched in the
*Fusarium* wilt rhizosphere and protect cucumber from pathogen infection by
increasing the level of reactive oxygen species in the roots ⁷. Similarly, the *Fusarium*-
infected cabbage recruited beneficial *Pseudomonas* which significantly suppressed
pathogen growth and enhanced plant disease resistance ⁸. These findings indicate that
certain general changes may exist in the rhizosphere microbiome of diseased plants.
Understanding such general changes will provide deep insights into the recruitment
mechanism of rhizosphere bacterial communities".

**We believe this addition provides a more balanced perspective on plant-**
**pathogen-bacteria interactions. Thank you again for your constructive**
**comments.**

(5) In addition, I am extremely hesitant to accept the term 'robust variation'. It might
lead to confusion.

**Answer: Corrected.**

**We agree that the term "robust variation" could potentially lead to confusion.**
**As suggested, we have revised the term to "general variation" to ensure clarity**
**and avoid the misinterpretation.**

(6) Specific comments: Abstract: Line 30. What of the strain was stimulated? The
growth? It would be appropriate to clarify it.

**Answer: Corrected.**

**We have revised the text to clarify that the growth of *F. anhuiense* was**
**significantly stimulated by tocopherol acetate. The updated sentence now reads:**

"The growth of *F. anhuiense* was significantly stimulated by tocopherol acetate."

(7) Introduction:

Generally, the introduction covers the background and motivation for conducting this
study. Nevertheless, in my opinion, the syntax should be improved, as many sentences
are imprecise or misleading, furthermore, the text seem to lack a logical flow.

**Answer: Corrected.**

**We have carefully revised the syntax and improved the clarity and precision of**
**the text. Additionally, the revised manuscript has been professionally edited by**
**AJE (American Journal Experts) to ensure high-quality language standards.**

**We have also restructured the introduction to enhance its logical flow, taking**
**into account your valuable suggestions. We believe these changes have**
**significantly improved the readability and coherence of this section.**

(8) To give some examples Line 40. "Beneficial bacteria can be enriched in the
rhizosphere of plants infected with pathogens". I think it is reasonable to first
introduce the concept of dysbiosis (e.g. what described from line to 48) and then
describe that among a dysbiosis certain changes could be beneficial for the host, while
other could also be negative.

**Answer: Thank you for your suggestion. We have revised the text. Please see**
**suggestion 4 (line 540).**

(9) Line 49. "The microbial recruitment mechanism is the key theme in microbial
ecology". This sentence sounds odd and misleading, I am sure microbial recruitment
mechanisms are a very important topic of research of microbial ecology, but not the
only key topic.

**Answer: Regarding point (9), we agree that the original statement on Line 49**
**was overly broad and could be misleading. As suggested, we have revised the**
**sentence to better reflect the importance of microbial recruitment mechanisms**

**within the broader context of microbial ecology. The corrected sentence now**
**reads:**

"Microbial recruitment mechanisms are an important topic in microbial ecology."

(10) Line 50. Briefly explain why these tools can help understand recruitment
mechanisms

**Answer: Corrected.**

**Based on your suggestion 7, we have reorganized this section. We first describe**
**how rhizosphere microbial assembly is driven by root exudates, followed by the**
**impact of pathogen invasion on root exudate composition. These altered root**
**exudates can recruit beneficial microbes to combat pathogen invasion.**
**Additionally, pathogen invasion affects rhizosphere microbiome functions, and**
**changes in nutrient-related genes reflect shifts in microbial composition. Finally,**
**we conclude that integrating metabolomics and metagenomics provides a**
**systematic approach to unraveling microbial recruitment mechanisms.**

**For more detailed information, please refer to our response to Reviewer 1,**
**Question 2, line 87.**

(11) Results: Line 85. The authors used operational analysis units (OAUs), which is a
valid approach, but, in my opinion, the grouping criteria should be specified either
here or in the methods section, as it is not clear how these groups were formed.

**Answer: Corrected.**

**We have revised the text to clarify the concept and grouping criteria of OAUs.**
**The updated explanation is as follows:**

"Within a single study from our meta-dataset, variations in sampling locations and
665 times can significantly disrupt the comparison between healthy and diseased
rhizosphere microbiomes. To address this, we introduced the concept of Operational
Analysis Units (OAUs), which refine the analysis by subdividing samples into
subgroups until comparisons between diseased and healthy rhizosphere microbiomes
can be made under the same conditions (e.g., same sampling location and time within
a study). Samples from the same sampling location or time are grouped into one OAU.
These subgroups are defined as OAUs."

**For more detailed information about the criteria used to form OAUs, please**
**refer to our response to Reviewer 1, Question 3, line 127.**

(12) Line 87 to 93: The authors found a total of 37 studies where Fusarium was
analyzed, 7 studies about Fungi, 6 having low quality, 9 lacking metadata and 2

adding bacterial wilt, were also removed. Now, $37 - (7 + 6 + 9 + 2) = 13$; but in the
text it says that you only retained 9 studies. Am I missing something, or there is an
error in the text?

**Answer:**

**We have revised the sentence to clarify the exclusion process and ensure the**
**calculations are accurate. The updated text now reads:**

"Six studies for other diseases were excluded (2 for *Ralstonia solanacearum* bacterial
wilt, 2 for nematode diseases, and 2 for fungal diseases caused by *Rhizoctonia* and
*Cylindrocarpon*)."

**With this correction, the calculation is as follows:**

**Total studies initially identified: 37**

**Excluded studies:**

**7 studies about Fungi**

**6 studies with low quality**

**9 studies lacking metadata**

**6 studies for other diseases (2 bacterial wilt, 2 nematode diseases, 2**
**fungal diseases)**

**Total excluded: $7 + 6 + 9 + 6 = 28$**

**Retained studies: $37 - 28 = 9$**

(13) Line 89. Explain the reasons to remove the fungal amplicon studies.

**Answer: We have revised the text on Line 89 to provide a clearer explanation.**
**The corrected sentence now reads:**

"Studies investigating changes in the fungal community (7 amplicon studies) were
excluded from the analysis, as most of them lack sufficient metadata information."

(14) Line 109. What about observed richness? It is the fundamental index and the
others rely on it

**Answer: We appreciate your suggestion to include this fundamental index.**

**We have revised the text on Line 109 to incorporate the findings related to**
**observed richness. The corrected sentence now reads:**

"However, there was a modest increase in observed richness in the bacterial

community of the diseased rhizosphere ($P < 0.001$) (Fig. S1)"

(15) Line 114. It appears to me that there are more appropriate statistical techniques to
evaluate the effect of various predictors on community composition, such as the use
of PERMANOVA with a strata argument. Why did the authors prefer to the PCoA
axis and mixed effect models? If previously the effects of OAU were assessed with
PERMANOVA, why not for infection status? That would allow a direct comparison
of the explanatory power (R^2).

**Answer: Thank you for your insightful comment regarding the statistical**
**techniques used in our analysis. We appreciate your suggestion to use**
**PERMANOVA with a strata argument for evaluating the effect of predictors on**
**community composition. However, we chose to use Principal Coordinate**
**Analysis (PCoA) and Generalized Linear Mixed Models (GLMM) for the**
**following reasons:**

1. **Handling Environmental Variability:** Rhizosphere soil is an open
environment where factors such as soil type and plant type can significantly
influence the microbiome. These environmental factors can obscure the effects
of pathogen invasion. GLMM is particularly suited for such scenarios because
it can account for both fixed effects (e.g., infection status) and random effects
(e.g., soil type, plant type), making it a robust choice for meta-analysis.
- 2. **Meta-Analysis Context:** GLMM is widely used in meta-analyses to integrate
data from multiple studies with varying experimental conditions. It allows us
to control for heterogeneity across studies and isolate the specific effects of
pathogen invasion on the rhizosphere microbiome.
- 3. **Supporting Evidence:** To complement the GLMM results, we also analyzed
the differences between diseased and healthy rhizosphere microbiomes within
the same environment (as shown in Fig. 2). This approach confirmed that
pathogen invasion obviously alters the rhizosphere microbiome in 14 out of 25
OAU, consistent with the GLMM findings.
- 4. **Consideration of PERMANOVA Limitations:** Based on the suggestion of
Reviewer 1, we performed a multivariate homogeneity of group dispersions
test using the "betadisper" function in *R* to assess the assumption of
homogeneity of variances. The result of the betadisper test was significant (P
= 0.001), indicating that the observed differences might indeed be influenced
by variations in dispersion. To ensure robust and reliable results, we have
removed the PERMANOVA results from the manuscript.

While PERMANOVA is a powerful tool for assessing community composition

differences, it does not inherently account for random effects, which are critical in our
analysis, and its interpretation can be limited by heterogeneity in group dispersions.
By using GLMM, we aimed to provide a more comprehensive and nuanced
understanding of the impact of pathogen invasion across diverse environmental
conditions.

**We hope this explanation clarifies our rationale for choosing these statistical**
**methods. Thank you again for your valuable feedback, which has helped us**
**refine our manuscript.**

(16) Line 126. Is the random forest approach including OAU. If it is not I would
reconsider the approach. OAU explains 76% of the variability, not including it in
further analyses is a critical flaw

**Answer: Thank you for your comment regarding the inclusion of Operational**
**Analysis Units (OAUs) in the random forest approach. We appreciate your**
**concern and would like to clarify our rationale for not including OAUs in this**
**analysis.**

1. **Purpose of Random Forest Analysis:** The random forest approach was used
to identify biomarker microbes that distinguish between healthy and diseased
samples. The input data for this analysis consists of predictor factors (e.g.,
microbial taxa) and their abundances, along with the sample group labels
(healthy or diseased). OAUs, which represent subgroups based on sampling
locations and times, are not suitable as predictor variables because they do not
directly differentiate between healthy and diseased samples. Instead, OAUs
contain both healthy and diseased samples within the same subgroup.

Table1 The example data format for random forest approach

Taxa1	Taxa2	Taxa3	Sample group
1	1.1	1.4	Dis
2	1.2	1.5	Dis
3	1.3	1.6	Dis
10	10.1	13.1	Hea
11	11.1	14.1	Hea
12	12.1	15.1	Hea

2. **OAU as a Random Effect:** OAUs account for random effects such as
sampling time and location, which influence the rhizosphere microbiome.
However, the goal of the meta-analysis was to isolate the specific effects of
pathogen invasion on the microbiome by removing the interference of these
random effects.

3. **PERMANOVA Results:** The statistic R (76%) corresponds to an R^2 value of

57.6%, which reflects the variability explained by OAUs. In addition, as noted
in our response to suggestion 15, the significant result of the betadisper test (P
= 0.001) indicated that the observed differences might be influenced by
variations in dispersion. Consequently, we removed the PERMANOVA
results from the manuscript to ensure robust and reliable conclusions.

**In summary, the random forest approach was designed to focus on identifying**
**microbial biomarkers that differentiate healthy and diseased samples,**
**independent of the random effects accounted for by OAUs. For more detailed**
**information about the criteria used to form OAUs, please refer to our response**
**to Reviewer 1, Question 3, line 127.**

**We hope this explanation addresses your concern and clarifies our methodology.**

(17) Line 136. (a) What do you mean by “antioxidant catalog”? Is there a gene catalog
for antioxidant genes you used? Or did you choose a specific KEGG group? Can you
provide accession numbers in case? (b) Line 135 to 148. Please provide references
that link genes to functions. Moreover, in the methods you say that you grouped genes
into orthologs, but here you use the term genes. Are you talking about KEGG
orthologs or genes? Please clarify and in case you are talking about orthologs, add the
KO accession number.

**Answer: Thank you for your detailed comments regarding the functional**
**analysis of the microbiome. We have revised the text to clarify the use of KEGG**
**orthologs and provided additional details, including KO accession numbers and**
**references linking genes to functions. Below is the updated section:**

**Revised Text:**

"We next determined the general changes in the microbiome functions of the diseased
rhizosphere (Table 1 and Table 2). A KEGG ortholog (a set of orthologous genes that
are functionally conserved across different organisms) encoding thioredoxin-
dependent peroxiredoxin (K03564), associated with antioxidant functions (Table S6),
was markedly enriched in the diseased rhizosphere. Similarly, KEGG orthologs
involved in sulfur metabolism, including tRNA-uridine 2-sulfurtransferase (K00566),
sulfur-carrier protein (K21147), and sulfur-carrier protein adenylyltransferase (K21147),
also showed increased abundance in the diseased rhizosphere. In contrast, KEGG
orthologs linked to root exudate metabolism, such as malate dehydrogenase (K00029),
fumarate hydratase (K01679), and dihydroxy-acid dehydratase (K01687), were more
abundant in the healthy rhizosphere. Notably, KEGG orthologs associated with chitin
degradation, including acetylhexosaminidase (K01207), were highly enriched in the
diseased rhizosphere. Furthermore, KEGG orthologs related to stress responses, such
as integrase (K04763), phosphate regulon sensor histidine kinase (K07636), and
NAD(P)H-hydrate dehydratase (K17759), were also elevated in the diseased

rhizosphere. Moreover, several of these KEGG orthologs (e.g., K01207,
acetylhexosaminidase; K04763, integrase; and K07636, phosphate regulon sensor
histidine kinase) were identified in the *F. anhuiense* K5 genome (Table S7). "

**Additional Clarifications:**

KEGG Orthologs vs. Genes: We used KEGG orthologs (KO) rather than individual
genes. The term "catalog" refers to a group of KEGG orthologs with similar functions.

Accession Numbers: The KO accession numbers for the mentioned orthologs have
been added to the revised text.

References Linking Genes to Functions: References supporting the functional
annotations of these KEGG orthologs are provided in Table S6 (Supplementary
Materials).

**We hope these revisions address your concerns and provide greater clarity**
**regarding the functional analysis.**

(18) Discussion: Line 200. I am somehow not understanding the logical link between
how the listed gene categories changing contribute to positive feedback for the host
health. I understand degradation of chitin, but why would you expect lower
degradation on fumarate and malate in infected communities? Could you clarify in the
text why you expect these functions to be relevant in host/microbiome interactions?

**Answer: Thank you for your comment regarding the logical link between the**
**changes in gene categories and their contribution to positive feedback for host**
**health.**

**Upon reflection, we acknowledge that the connection between the observed**
**changes in fumarate and malate degradation and their relevance to**
**host/microbiome interactions was not sufficiently supported by published**
**literature. To address this, we have removed the speculative statements about the**
**relevance of these functions to host/microbiome interactions.**

(19) Line 205. "to achieve this goal, we performed...", you already said in the
discussion that you performed a metanalysis. No need to repeat it once again, I think
at this point of the paper it is clear to the reader.

**Answer: Corrected.**

**We have removed the redundant sentence on Line 205.**

(20) Line 218. I do not understand why you speak about functions both between lines
193 to 202, then you start speaking about Flavobacterium, and then you go back

speaking about functions again; as for the introduction, I would advise to restructure
the text.

**Answer: We have restructured the text to ensure a more coherent and logical**
**presentation. Specifically:**

**In the discussion section, we have integrated the content related to microbiome**
**functions into a single, cohesive segment to avoid fragmentation.**

**In the introduction, we have carefully revised the logic and flow to ensure that**
**the ideas are presented in a clear and sequential manner.**

**Thank you again for your valuable input, which has helped us enhance the**
**overall quality of our work.**

(21) Line 218. What do you mean by “generalizing the microbiome functions”

**Answer: We acknowledge that the phrase "generalizing the microbiome**
**functions" was ambiguous and did not effectively convey our intended meaning.**
**As part of the broader restructuring of the text (as mentioned in our response to**
**your earlier suggestion²⁰), we have removed this sentence entirely.**

(22) Methods: I suggest adding a description of how OAU were determined to clarify
how data was grouped together.

**Answer: Corrected. Please refer to our response to suggestion 11, line 655.**

(23) Line 259. I think a list of the studies, including if they were used or not, should
be reported in supplementary information. I see that similar information is already
provided in supplementary table S1 and S2; nevertheless, I think it is still difficult to
understand which studies were originally filtered and which were kept, and the
corresponding publication.

**Answer: To address this, we have created a new Table S1 in the supplementary**
**materials to include a complete list of all studies considered, along with their**
**corresponding publications and whether they were included or excluded in the**
**final analysis. The new table provides the following details:**

1. **Study Identification:** The name or identifier of each study.

2. **Publication Reference:** The corresponding publication for each study.

3. **Inclusion/Exclusion Status:** Whether the study was included in the final
analysis or excluded, along with the reason for exclusion (e.g., low quality,
lack of metadata, or focus on non-target pathogens).

**This revision ensures that readers can easily track which studies were originally**
**filtered, which were retained, and the rationale behind these decisions.**

**We hope this addition addresses your concern and enhances the clarity and**
**transparency of our manuscript.**

(24) Line 262 to 265. Put the complete search terms in quotation marks

**Answer: Corrected.**

(25) Line 295: It is unclear to me what “which were then predicted for overlapping
groups with Prokka”. Prokka is an annotation tool, what does it mean, it predicts
overlapping groups? Can it be that you mean “open reading frames”, or “gene coding
sequences”?

**Answer: We agree that the original sentence on Line 295 was ambiguous. As**
**suggested, we have revised the sentence to clearly describe the process. The**
**corrected text now reads:**

"The assembled sequence was predicted for open reading frames via Prodigal v2.6.3."

(26) Line 303. lmer is not a function of the package lmerTest

**Answer: Corrected.**

**We have corrected the sentence on Line 303 to accurately reflect the use of the**
**lmer function from the lme4 package. The revised text now reads:**

"The combined analysis, which was based on a previous study, was conducted via a
linear mixed-effects model via the lmer function of the lme4 package."

(27) Line 307. adonis is the vegan function, not the method

**Answer: Corrected.**

**The corrected text now reads:**

"To determine the significance in difference of community composition between
healthy and diseased samples, a similarity analysis (ANOSIM) was performed using
the anosim function in vegan package."

(28) Line 310. I am not an expert in reporting pot experiments, but it seems to me that
the description is lacking details important for reproducibility. For example, how old
were the seedlings? Were they grown with the pathogen for only 10 days? How many
978 ml for how much soil?

**Answer: We have revised the text on Line 310 to provide a more detailed**
**description of the experimental setup. The corrected text now reads:**

"Five tomato seedlings were planted in the soil from Nanjing as described above. The
tomato planting process was similar to the previous greenhouse experiment, with each
pot containing 0.5 kilograms of soil. A 20 ml pathogen suspension (approximately $1 \times$
10^6 CFU ml⁻¹) was inoculated into the root-stem junction of 30-day-old seedlings."

**Thank you again for your constructive feedback.**

(29) Line 324. Was the genome sequenced by you? There are no details regarding the
sequencing. How was it conducted? . On which platform? I suppose you used short
reads by the assembler you used, but I should not guess, it should be specified.

**Answer: We have revised the text on Line 324 to provide a detailed description**
**of the genome sequencing methodology. The corrected text now reads:**

"Genomic DNA was extracted by using the cetyltrimethylammonium bromide (CTAB)
method with minor modifications, and then the DNA concentration, quality and
integrity were determined by using a Qubit Fluorometer (Invitrogen, USA) and a
NanoDrop Spectrophotometer (Thermo Scientific, USA). Sequencing libraries were
generated using the TruSeq DNA Sample Preparation Kit (Illumina, USA) and the
Template Prep Kit (PacificBiosciences, USA). Genome sequencing was then
performed by Personal Biotechnology Company (Shanghai, China) by using the
Illumina Novaseq platform."

(30) Line 343: Please report which studies were reviewed.

**Answer: Corrected.**

**We have added the details of the reviewed studies to Table S8 in the**
**supplementary materials. This table now provides a comprehensive list of the**
**studies included in our analysis, ensuring that readers can easily access this**
**information.**

(31) Line 358. How were these root exudates made? In vitro, or purchased? Is the
description on the original paper you quote?

**Answer:**

**The root exudates used in our experiments were purchased from chemical**
**reagent companies. The descriptions of these root exudates are consistent with**
**those reported in previously published papers.**

(32) Line 375. How was the sequencing made? Which platform?

**Answer: We have revised the text on Line 375 to provide a clear description of**
**the sequencing process. The corrected text now reads:**

"Total genomic DNA from 0.5g samples was extracted using the FastDNA SPIN Kit
for Soil (MP Biomedicals, CA). The bacterial 16S rRNA V3-V4 region (F: 5'-
ACTCCTACGGGAGGCAGCA-3'; R: 5'-GGACTACHVGGGTWTCTAAT-3') was
sequenced on an Illumina Novaseq 6000 (Illumina, San Diego, CA, USA) provided
by Personalbio Technology Co. Ltd. (Shanghai, China)."

**(33) Line 428. Which data is normally distributed?**

**Answer: We have revised the text on Line 428 to explicitly state which data were**
**normally distributed. The corrected text now reads:**

"The following data followed a normal distribution according to the Shapiro–Wilk
test. The cell density (OD₆₀₀) of the *F. anhuiense* K5 strain in vitro, the abundances of
the root exudates, the Shannon index values, and the expression levels of genes
related to TonB-dependent transporters and the major facilitator superfamily were
normally distributed and assessed via the two-tailed Student's t test".

**We hope this revision addresses your concern and provides a clearer explanation**
**of the statistical analysis.**

**(34) Figures: Figure 1. Improve description, what does the top right p-value indicate?**

**Answer: Corrected. The top right p-value was from the results of Generalized**
**Linear Mixed Model.**

**(35) Supplementary Fig. 2 should be part of Fig. 1 to show the reader the large effect**
**of OAU. Any stats to support the statement of 'clearly distinct clusters'?**

**Answer: Regarding point (35)**

**(a) We acknowledge that the significant result of the betadisper test ($P = 0.001$)**
**indicates that the observed differences might be influenced by variations in**
**dispersion. To ensure the reliability of our findings, we have removed these**
**results from the manuscript.**

**(b) To support the statement of "clearly distinct clusters" we have referenced the**
**ANOSIM results, which are provided in Table S4. This analysis confirms the**
**statistical significance of the observed clustering patterns.**

**We hope these revisions address your concerns and enhance the clarity and**
**robustness of our manuscript.**

(36) Figure 2A. Please correct x-axis label. Also what relative abundance is
represented? Mean across all studies? Differences between healthy and infected?
Please clarify, it is unclear.

**Answer: The x-axis label has been corrected. The relative abundance values**
**shown in Figure 2A represent the mean across all studies included in the analysis.**

(37) Figure 3 is really hard to interpret. I would suggest transforming it into a table
with full name, KEGG category and accession numbers, direction, intensity and
statistical significance of the change (or some other visualization approach).

**Answer: Corrected.**

**We have transformed the data from Figure 3 into a table format, which includes**
**the full gene names, KEGG categories, accession numbers, direction and**
**intensity of changes, and statistical significance. This information is now**
**provided in Table 1, making it easier for readers to interpret the results.**

**Reviewers' comments:**

Reviewer #1 (Remarks to the Author):

**The author's response is sufficient, and I have no further comments to add.**

**Answer: Thank you for your positive feedback. We appreciate your time and**
**thoughtful comments on our manuscript.**

Reviewer #2 (Remarks to the Author):

**(1) The authors showed many effects in the revised manuscript, such as adding new**
**metabolite experiments. However, there is no in planta assay to validate the findings.**
**The authors only suggested a proposal concept in Fig. S6. Nature Communications is a**
**high-ranking journal, and I believe the keystone taxa's result in planta assay is necessary.**

**Answer: We sincerely appreciate the reviewer's insightful comments and**
**constructive suggestions. We fully agree that in planta validation is critical to**
**strengthen the biological relevance of our findings.**

**As suggested, we have now performed additional in planta experiments to**
**validate the results of keystone taxa using a hydroponic co-cultivation system: (1)**
**Tocopheryl acetate significantly stimulated the growth of *Flavobacterium*. (2)**
***Flavobacterium* exhibits significant pathogen-suppressing effects both in the**
**presence and absence of tocopheryl acetate. (3) *F. oxysporum* was significantly**
**stimulated by tocopherol acetate.**

**These results demonstrate the complex plant-microbiota interactions**
**occurring during pathogen challenge. Most importantly, our data confirm that *F.***
***anhuiense* K5 maintains robust suppression of *F. oxysporum* in planta, and that**
**tocopheryl acetate - though released during pathogen invasion - does not**
**significantly alter pathogen proliferation outcomes in our experimental system**
**when *Flavobacterium* is present.**

**These in planta findings directly validate our previous results, providing clear**
**evidence that: (1) *Flavobacterium* possesses intrinsic pathogen-inhibiting capacity.**
**(2) Tocopheryl acetate serves as a key growth stimulant for *Flavobacterium*.**

**The reviewer's expertise has been invaluable in strengthening our work, and**
**we sincerely appreciate their thoughtful contribution to improving this study.**

**These results have been incorporated in the revised manuscript:**

**Results part (line 268 -283)**

**Inhibitory effect of *F. anhuiense* against *F. oxysporum* and *F. anhuiense***
**population enhancement by tocopherol acetate in planta**

We investigated the effects of *F. anhuiense* against *F. oxysporum* and effects of
tocopherol acetate on both strains at 24 and 72 h in planta. Quantitative analysis
revealed that *F. anhuiense* treatment significantly reduced *F. oxysporum* gene copy
numbers by 4.97-fold compared to the control at 24 h (Fig. 6). This suppressive effect
persisted in the presence of tocopherol acetate, with the combined treatment (tocopherol
acetate + *F. anhuiense*) showing a 4.62-fold reduction in *F. oxysporum* gene copies
relative to tocopherol acetate treatment alone. However, the combined treatment
showed no statistically significant reduction in the copy numbers of *F. oxysporum*-
related gene compared to the control. Moreover, tocopherol acetate significantly
increased the gene copy numbers of both strains, with *F. anhuiense* gene copies showed
a 3.16-fold increase in the combined treatment compared to *F. anhuiense* alone, while
*F. oxysporum* gene copies exhibited a 3.17-fold increase in tocopherol acetate treatment
versus *F. oxysporum* alone. These trends remained consistent at 72 h (Fig. S10),
demonstrating temporal stability in the observed interactions.

**Discussion part (line 354 - 365)**

Moreover, the growth of *F. oxysporum* was stimulated by tocopherol acetate. While
tocopherol acetate has been shown to significantly inhibit fumonisin production in
*Fusarium verticillioides*³⁵. It suggests a potential trade-off between secondary
metabolite production and cellular growth³⁶. The co-enrichment of *F. oxysporum* and
*Flavobacterium* reflects the intricate interplay between plants and their associated
microbiota. *F. anhuiense* K5 consistently suppressed *F. oxysporum* growth in planta,
regardless of tocopheryl acetate presence. Importantly, the enrichment levels of *F.*
*oxysporum* in combined treatment (tocopherol acetate + *F. anhuiense* K5) showed no
significant difference compared to pathogen-only controls (Fig. 6A). These results
collectively indicate that although tocopherol acetate is released by plants following
pathogen invasion, this response does not significantly promote pathogen proliferation
under the experimental conditions tested.

**We sincerely appreciate the reviewer's valuable insights again, which have**
**significantly strengthened our study.**

Reviewer #5 (Remarks to the Author):

This is a revised version of a manuscript previously submitted to Nature
Communications. I was not involved in the first round of review, so my assessment
focuses primarily on whether the authors have adequately addressed the reviewers'
previous comments.

Overall, I find that the manuscript has been significantly improved in this revised
version. However, my main concern is related to the availability of the code. The
authors have provided R scripts via a Zenodo repository, but they have not included the
input data or sufficient documentation to explain how to run the scripts. These are
essential components for ensuring reproducibility and should be addressed before
publication.

**Answer: We sincerely appreciate the reviewer's valuable suggestion regarding**
**code availability and reproducibility. In response, we have thoroughly updated**
**our code repository to include:**

**(a) All necessary input data required to execute the scripts.**

**(b) Detailed documentation (README files) explaining script execution and**
**dependencies.**

**(c) Version information to ensure compatibility and transparency.**

**The enhanced repository, now available at <https://zenodo.org/records/16676212>,**
**contains all necessary components to exactly reproduce our analyses.**

Reviewer #6

(Remarks to the Author)

The paper “General variation in the *Fusarium* wilt rhizosphere microbiome” by Su et
al presents a meta-analysis and experimental validation on how *Fusarium* wilt infection
affects the rhizosphere bacterial community and root exudates. It focuses particularly
on: Changes in bacterial community composition and function. Role of root exudates
(especially tocopherol acetate) Isolation and behavior of *Flavobacterium* strains, which
are identified as the culprit in *Fusarium* wilt rhizosphere microbiomes. Identifying
*Flavobacterium* and in particular *F. anhuiense* as significant component of *Fusarium*
wilt and Tocopherol acetate as a growth stimulant are the central findings.

These are novel findings and not trivial, as tocopherol isn't known to be a common
microbial growth stimulant.

**Answer: We sincerely appreciate the reviewer's thoughtful evaluation of our work**
**and their recognition of the novelty of our findings. Thank you again for this**
**constructive engagement with our study's central contributions.**

Major revisions:

(1) The observed diversity increase in disease samples seems counterintuitive, as
dysbiosis commonly is characterized by diversity decrease. As such it would be great
if the authors could elaborate on this counterintuitive circumstance, at the very least in
the Discussion section.

**Answer: We sincerely appreciate the reviewer's insightful observation regarding**
**the counterintuitive increase in bacterial diversity in disease samples. Indeed,**
**dysbiosis is often associated with reduced microbial diversity, but the impact of**
**pathogen invasion on soil microbial communities remains context-dependent, with**
**studies reporting both increases and decreases in diversity.**

**Below, we elaborate on the potential possibility underlying this phenomenon:**
**The direction of diversity shifts likely depends on the environment, soil**
**microbiome and pathogen species. For example, in low-pH soils, *Fusarium* may**
**produce bikaverin, which selectively inhibits bacteria like *Bacillus* and**
***Actinomyces*, potentially decreasing diversity. Whereas pathogens with limited**
**antibacterial effects may instead foster bacterial diversification through**
**saprophytic activities (e.g., lignin degradation) that release novel nutrients.**

**We have expanded our Discussion to better address this complexity. Please**
**see line 293**

*Fusarium* infection could significantly affect alpha bacterial diversity in our study
(Fig. S6). However, the impact of *Fusarium* infection on bacterial alpha diversity

remains complex, with studies reporting both increases and decreases in the diversities
155 ^{21,22}. These contrasting effects likely depend on environmental conditions or native soil
microbial composition. For instance, in low-pH soils, *Fusarium* may produce bikaverin,
a metabolite that selectively inhibits certain bacteria (e.g., *Bacillus* and *Actinomycetes*),
potentially reducing diversity ²³. Conversely, when pathogen activity involves
primarily saprophytic functions (e.g., lignin degradation) without strong antibacterial
effects, it may indirectly promote bacterial fitness or diversity by releasing
supplemental nutrient resources ²⁴.

(2) For presentational purposes, it would be desirable to provide visualizations that
allow the inspection of the microbial community composition on various taxonomic
ranks QIIME2 for example provides web based tools for this matter (stacked bar
diagrams). It allows the sorting of samples based on particular genera or “species” of
interest (in this case: the top predictors for diseased classes like *Flavobacterium*,
*Variovorax*, *Stenotrophomonas* etc.).

**Answer: We sincerely appreciate the reviewers' insightful comments regarding**
**the visualization of microbial community composition.**

**In response, we have provided stacked bar plots displaying taxonomic**
**profiles at the phylum, order, and genus levels (Fig. S4). While QIIME2 View**
**(<https://view.qiime2.org>) offers useful interactive features, we opted to generate**
**publication-quality figures using ggplot2 due to resolution limitations in the web-**
**based tool's exports.**

**We would like to clarify that while QIIME2 View allows sample sorting based**
**on metadata features, it may do not currently support direct sorting by specific**
**taxonomic groups. This is because in the input data structure (as shown below),**
**taxonomic information and metadata features are integrated in the same row**
**format:**

Table1 The example data format for stacked bar diagrams in QIIME2 View

Sample	TAXA1	TAXA2	TAXA3	Feature1	Feature2	Feature3
S1	0	0	0	F1	FF1	FFF1
S2	0	0	0	F2	FF2	FFF2
S3	0	0	104	F3	FF3	FFF3
S4	0	0	50	F1	FF1	FFF1
S5	0	0	47	F2	FF2	FFF2

**We observed significant enrichment of disease-associated taxa (e.g.,**
***Flavobacterium*, *Variovorax*, and *Stenotrophomonas*) in the diseased rhizosphere**
**samples. To better visualize these patterns, we implemented sample grouping by**
**health status (healthy vs. diseased) using ggplot2's facet functionality. In addition,**
**the relative abundances of top predictors were shown in box plots (Fig. 3).**

**We appreciate your insightful comments and hope these visualizations clarify our**
**findings.**

(3) Likewise, I would have expected a standard visualization of all alpha diversity
scores, e.g. as box plots. None of the used alpha diversity measures take phylogenetic
distance into account. It would be desirable to include at least one, like Faith's
phylogenetic distance.

**Answer: We sincerely appreciate the reviewer's suggestion regarding alpha**
**diversity analysis.**

**As recommended, we have calculated and visualized all major alpha diversity**
**indices (including Shannon, Observed features, Chao1, ACE, Simpson, and**
**Faith's PD). These results are presented in Fig. S1 of the revised manuscript. The**
**results showed that the Faith's PD diversity were significantly increased in the**
**diseased rhizosphere.**

**We have incorporated the content into the revised manuscript. Please see the**
**follows: line 133**

We calculated common metrics for alpha diversity. No significant or consistent
changes in Shannon ($P=0.06$) and Simpson ($P=0.84$) diversity were observed between
diseased and healthy rhizospheres via the linear mixed effects model. However, there
was a modest increase in observed richness, ACE, Chao, Faith phylogenetic diversity
indices in the bacterial community of the diseased rhizosphere ($P<0.001$) (Fig. S1).

**We thank the reviewer for this valuable comment, which has helped improve the**
**clarity of our diversity analysis.**

(4) Finally, it is very common and useful to support microbiome analysis with beta-
diversity analysis. The authors perform PCoA, which is a rather technical term and
beta-diversity should be used where possible. The details of beta-diversity are not well-
described. The code for Figure 1 seems to use Bray-Curtis, but this is not described in
the manuscript/Methods section.

**Answer: We sincerely appreciate the reviewer's valuable suggestions regarding**
**beta-diversity analysis. In response to these comments, we have made the**
**following improvements:**

(a) **We have replaced the technical term "PCoA" with "beta-diversity"**
**throughout the manuscript where appropriate.**

(b) **We have implemented UniFrac distance metrics (weighted) for our beta-**
**diversity analysis at ASV level. This change better accounts for phylogenetic**
**relationships among microbial taxa. Our results were consistent with those**

observed at the genus level (Fig. S1 and S3). We have thoroughly documented
these analytical approaches in the Methods section (line 431).

**Please see the followings**

Beta-diversity was assessed by computing: (1) Bray-Curtis dissimilarity based on
genus-level abundance profiles, and (2) weighted UniFrac distance matrices derived
from ASV-level phylogenetic data, using the vegan package⁴³ in R.

(5) UniFrac[1] is recommendable as it takes phylogenetic distances of community
constituents into account. The provided code for Figure 1 seems to use the Bray-Curtis
distance. Various visualization methods like a clustered heatmap with hierarchical
clustering dendrograms can give a formidable overview over all samples and how they
relate to each other.

**Answer: Following the recommendation, we have implemented weighted UniFrac**
**as our distance metric at the ASV level, which better captures phylogenetic**
**relationships among microbial taxa. We initially generated clustered heatmaps**
**with hierarchical clustering dendrograms as suggested.**

**We found PCoA plots provided better visual separation between healthy and**
**diseased samples. The patterns of microbial community differences were more**
**readily interpretable in this format. To ensure transparency and completeness:**
**The heatmap visualizations have been included as Fig. S2.**

**These modifications have significantly strengthened our analytical approach**
**while maintaining clarity in data presentation. We are grateful for these valuable**
**suggestions that have enhanced our study's methodological rigor.**

(6) A particular issue is that the partitioning of the data into 25 “operational analysis
units” (OAUs), many of them containing below 10 data points (some have only 6),
reduces statistical power. 6 samples in a PCoA cluster easily by chance. It is also not
clear according to which criteria the OAUs are defined.

**Answer: (a) We appreciate the opportunity to clarify the concept of OAUs.**

**When performing combined analyses across different studies, random effects**
**such as sampling locations, sampling times, and other confounding factors can**
**obscure the differences between control and treatment groups.**

**To address this issue, we employed the Generalized Linear Mixed Model**
**(GLMM), a widely used statistical model in meta-analyses that accounts for both**
**fixed effects (e.g., diseased vs. healthy plants) and random effects (e.g., sampling**
**locations, sampling times). For instance, Bisanz et al. (2019) used GLMM to**

analyze the effects of high- and low-fat diets on the gut microbiome across studies,
treating diet as a fixed effect and individual studies as random effects.

However, soil is an open environment where the microbiome is highly
sensitive to external factors. Within a single study, variations in sampling locations
and times can significantly disrupt the comparison between healthy and diseased
rhizosphere microbiomes.

To address this issue, we introduced the concept of Operational Analysis
Units (OAUs), which refine the analysis by subdividing samples into subgroups
until comparisons between diseased and healthy rhizosphere microbiomes can be
made under the same conditions (e.g., same sampling location and time within a
study). Samples from the same sampling location or time are grouped into one
OAU. These subgroups are defined as OAUs (see Fig. S11). This approach
provides a more intuitive way to compare diseased and healthy microbiomes
under controlled conditions.

We hope this explanation clarifies the rationale and methodology behind the
use of OAUs in our study. Thank you again for your valuable feedback.

**(b) As for less samples in OAU:**

We sincerely appreciate the reviewer's important observation regarding sample
sizes in our operational analysis units (OAUs). We fully acknowledge that smaller
OAU sample sizes (e.g., 6 samples) may reduce statistical power and increase the
risk of Type II errors. For cases like OAU24 (with 3 healthy and 3 diseased
samples), while the PCoA shows visual separation, we confirm this doesn't reach
statistical significance ($P > 0.05$), consistent with the reviewer's concern about
small sample sizes. Crucially, our mixed-effects modeling demonstrates that:

**When accounting for OAUs as random effects: Pathogen infection shows**
**highly significant effects on bacterial composition ($P < 0.001$)**

**Without this OAU-level control: The infection effect becomes nonsignificant**
**($P = 1$).**

**This confirms that the OAU framework, despite some loss of statistical power**
**in individual units, is essential for properly accounting for random effects.**

**To maintain analytical rigor in our OAU framework, we recognize this**
**approach may reduce statistical power in some units, potentially increasing Type**
**II error risk (false negatives wherein true effects are missed). Importantly, despite**
**this conservative methodology, our results still demonstrate statistically**
**significant impacts of pathogen invasion on bacterial community structure ($P <$**
**0.001).**

(7) Line 154: the text doesn't say what exactly the classification task is, ie. what are the
features, how many samples were used (was the learning done for each OAU
separately?) and what are the exact classes. The latter are likely to be healthy vs
diseased and the former the genera present in the respective sample but it's best to state
that explicitly and exactly in the Methods section.

**Answer: We sincerely appreciate the reviewer's suggestion for clarifying our**
**classification analysis. The machine learning task aimed to identify biomarkers in**
**the rhizosphere microbiome between healthy and diseased plants using microbial**
**community composition as features, with all bacterial genera's relative**
**abundances as predictors and binary health status as the response variable. This**
**analysis was performed globally across all 198 samples rather than within**
**individual OAUs, using random forest classification.**

(8) The choice of method to identify the best predictors is not well motivated. Random
Forests are common but far from the only method to identify predictors. First of all, it
should be demonstrated that the formulated Machine learning task is actually doing a
reasonable job, i.e. accuracy values should be reported. Generally, SHAP /TreeSHAP[2]
is currently the best practice to determine feature importance, as it associates high
explainability/interpretability with additive (and other desirable mathematical)
characteristics to features.

**Answer: We sincerely appreciate your insightful comments regarding the**
**methodology for identifying optimal predictors. You raise an excellent point about**
**the need for rigorous feature selection approaches. Here's how we addressed this**
**critical aspect:**

**(a) Initial Screening with Statistical Rigor**

Following established practices in machine learning (Lesmeister, *Mastering*
*Machine Learning with R*, Chapter 6), we initially performed microbial feature analysis
using the rfPermute R package. This method serves two key purposes: (1) effectively
filtering out noisy data, and (2) identifying statistically significant predictors based on
MeanDecreaseGini or MeanDecreaseAccuracy index (Type 1 predictors). Our
preliminary model demonstrated robust performance, achieving 70% accuracy.

**(b) Enhanced Predictive Performance**

To improve accuracy, we selected the top 20 statistically significant predictors
based on MeanDecreaseGini or MeanDecreaseAccuracy index (total 31 genera from
Type 1 predictors) and implemented automated hyperparameter tuning via the mlr
framework. This refined model achieved 80% accuracy (cross-validated),
demonstrating robust performance.

**(c) Incorporating SHAP Values (Per Your Suggestion)**

As you recommended, we have rigorously calculated the SHAP values (Top 10
features) and additionally cross-validated them with the model's built-in importance
factors (Top 10 features). Only those predictors that simultaneously met both criteria
were defined as Type 2 predictors to ensure the most robust feature selection.

374 **(d) Validation via Mixed Models**

All 31 type 1 predictors were re-evaluated using mixed-effects models to confirm
their associations with outcomes while accounting for covariates. Notably,
*Flavobacterium* (a Type 2 predictor) exhibited statistically significant effects in this
final validation step, reinforcing the robustness of this finding.

**We agree that SHAP values offer superior explainability, and we have highlighted**
**their role in our revised manuscript.**

383 **Please see the followings: line 154**

A random forest classifier was used to define the biomarkers of the bacterial response to *Fusarium*
infection (Fig. 2A). The top 20 most- predictive genera were selected based on the
MeanDecreaseGini or MeanDecreaseAccuracy indices. *Stenotrophomonas*, *Sphingobacterium*,
*Variovorax*, *Stella*, *Flavobacterium*, *Aeromonas*, *Luteimonas*, *Sporocytophaga*, *Novosphingobium*
were significant predictors in both the MeanDecreaseGini and MeanDecreaseAccuracy indices,
suggesting that the genera represent stable biomarkers.

To obtain stable and reliable biomarkers, we selected the top predictive genera that exhibited
significant statistical relevance in the MeanDecreaseGini or MeanDecreaseAccuracy index (31
genera in total) and performed another round of prediction using the automated hyperparameter-
tuning mlr machine learning framework with the random forest algorithm. The accuracy of this
prediction model increased by 10% compared to the initial predictive criteria, reaching 81% (Fig.
S5). The results demonstrated that *Stenotrophomonas*, *Sphingobacterium*, *Variovorax*, *Stella*,
*Flavobacterium*, *Luteimonas*, and *Sporocytophaga* were among the top 10 most predictive genera
(Table S6). This finding was consistent with the genera that showed statistical significance in both
the MeanDecreaseGini and MeanDecreaseAccuracy indices, confirming the robustness of our
selection methodology. SHAP (SHapley Additive exPlanations) analysis was employed to evaluate
feature importance. The results also identified *Stella*, *Flavobacterium*, *Sphingobacterium*, and
*Sporocytophaga* as the top predictive genera (Fig.2F). We further used a linear mixed effects model
to determine the significantly changed genera among these biomarkers. The results showed that
*Variovorax*, *Flavobacterium*, *Stenotrophomonas*, and *Novosphingobium* were significantly enriched
in the diseased rhizosphere. Moreover, *Flavobacterium* and *Stenotrophomonas*, and were the
potential beneficial bacteria that could inhibit the growth of *Fusarium* (Table S7). Overall,
*Flavobacterium* was the only genus consistently identified as a biomarker by multiple analytical
methods (Fig. 2B, Fig. 2F, Table S7), and it additionally exhibits potential biocontrol capabilities.

**Thank you for pushing us to strengthen this aspect.**

**(9) The reproducibility of the code is severely hampered:**

**(9.1) The version numbers for the required packages (most importantly: vegan) are not**

provided.

**Answer: We sincerely appreciate the reviewer's valuable suggestion regarding**
**code availability. The version numbers for the required packages were added in**
**the revived manuscript.**

(9.2) The code doesn't run out of the box, as it assumes the presence of certain objects
(pl in line 16, eam_gr in line 17 in Fig.1.R, marker_ok_u in) that are not instantiated.
I would assume that eam_gr contains some preprocessed microbial abundance data, but
that is not in the provided repository. The code would benefit from better commenting
and more intuitive variable names.

**Answer: We sincerely appreciate the reviewer's careful examination of our code**
**and their constructive feedback. To address these concerns and improve**
**reproducibility, we have made the following comprehensive updates to our code**
**repository:**

**Data Availability: We have now included all required input objects in the**
**repository.**

**Code Improvements: Created clear object documentation and user guidance**
**in the README file.**

**The enhanced repository, now available at <https://zenodo.org/records/16676212>,**
**contains all necessary components to exactly reproduce our analyses. We believe**
**these improvements will significantly enhance the usability of our computational**
**methods for the research community.**

(9.3) Non-standard libraries like lmerTest don't install via the install.packages
command. It is therefore important to provide installation instructions lmerTest.

**Answer: To ensure full reproducibility of our analyses, we have implemented the**
**following improvements:**

**Comprehensive Installation Documentation:**

Created a dedicated "INSTALLATION.R" file with detailed package installation
instructions.

For lmerTest, based on the documentation (<https://github.com/runehaubo/lmerTestR>),
you can use install.packages("lmerTest"). If this fails, alternatively use:

library("devtools")

install_github("runehaubo/lmerTestR")

(9.4) Given point 1-3. above, it is recommendable to provide a better environment for
code reproducibility, like a containerized version including all needed code, libraries
and ideally also data (or links/download instructions to (SRA) data repositories)

**Answer: We sincerely appreciate the reviewer's valuable suggestion regarding**
**code availability. We have now comprehensively improved our code repository to**
**ensure full reproducibility by:**

(a) adding detailed documentation including package versions and installation
instructions,

(b) providing automated SRA data download scripts, and

(c) enhancing code readability and usability through the addition of input data and
README guideline files.

**The enhanced repository, now available at <https://zenodo.org/records/16676212>,**
**contains all necessary components to exactly reproduce our analyses from raw**
**data to final results.**

**We believe these substantial improvements have fully addressed the**
**reproducibility concerns and significantly strengthened the computational**
**transparency of our study.**

Supplementary Material:

(10) Table S5: the table contains a column “References”, but doesn’t actually provide
complete references (no journal, author etc). The references are often also not referring
to seminal work, e.g.: Role of *Williamsia* and *Segniliparus* in human infections with
the approach taxonomy, cultivation, and identification methods is only cited 7 times.

**Answer: We sincerely appreciate the reviewer's insightful comments regarding**
**both the completeness and quality of our references. In response, we have**
**systematically improved Table S5 (renamed as Table S7 in revision) by providing**
**full reference details including all authors, complete journal names, publication**
**years. In addition, we have identified and incorporated the original publication**
**describing *Williamsia*, replacing the previous reference with this foundational**
**work.**

(11) The “Functions” column needs to be revised. It is out of my expertise, but a quick
check on *Williamsia* did not confirm it to be a well known “People pathogen”, possibly
it can be described as an opportunistic pathogen for humans.

**Answer: Corrected.**

(12) It would be very useful if the OAU tables also contained the number of samples
and the number of different studies these samples are coming from.

**Answer: Corrected. Please see Table S3.**

Minor revisions:

(13) Line 590 most likely contains a typo. GIMI should be Gini.

**Answer: Corrected.**

(14) The figures at the end (after line 845, page 42) are not numbered nor captioned.

**Answer: Corrected.**

(15) Table S4 Significant test of principal coordinate analysis in each OAU. Should be:
Table S4 Significance test ...

**Answer: Corrected.**

(Remarks on code availability)

(16) The reproducibility of the code is severely hampered:

The version numbers for the required packages (most importantly: vegan) are not
provided. The code doesn't run out of the box, as it assumes the presence of certain
objects (pl in line 16, eam_gr in line 17 in Fig.1.R, marker_ok_u in) that are not
instantiated. I would assume that eam_gr contains some preprocessed microbial
abundance data, but that is not in the provided repository. The code would benefit from
better commenting and more intuitive variable names. Non-standard libraries like
lmerTest don't install via the install.packages command. It is therefore important to
provide installation instructions lmerTest. Given point 1-3. above, it is recommendable
to provide a better environment for code reproducibility, like a containerized version
including all needed code, libraries and ideally also data (or links/download instructions
to (SRA) data repositories

**Answer: Please see the answers of suggestion 9.**

**Reviewers' comments:**

Reviewer #2 (Remarks to the Author):

Thank you for your revised submission, which incorporates the reviewers' suggestions.
I would like to provide the following comments regarding the newly added *in planta*
assay results.

**Answer:** Thank you very much for your thoughtful comments and suggestions
regarding our revised manuscript. We sincerely appreciate the time and effort you
have dedicated to help us improve our work. In response to your valuable
feedback, we have conducted additional *in planta* assays and incorporated the
following revisions:

In Figure 6, the pathogen density appears to be statistically reduced by treatment with
strain K5. However, I find three critical issues:

(1) The pathogen density remains above 10^6 copy numbers, which is still sufficient to
cause disease. Moreover, the copy number presented by the authors is not
pathologically meaningful. Results based on copy number should be converted into
cfu per gram. It is well known that *Fusarium* can occur disease development when
present at 10^5 cfu per gram.

**Answer:** We sincerely thank the reviewer for this insightful comment. The
reviewer pointed out that *Fusarium* can trigger disease development when present
at 10^5 cfu per gram. Indeed, as supported by previous studies (Li et al., 2018;
Matthews et al., 2023; Jambagi and Dixelius, 2023), *Fusarium* is capable of
causing disease at 10^5 CFU/g or 10^5 CFU/mL in soil or hydroponic system. Given
that *Fusarium* is a common pathogen in hydroponic tomato systems (Song et al.,
2004) and that hydroponic cultivation offers controllability and reproducibility
(Jambagi and Dixelius, 2023), we again selected the hydroponic system for
validation, as in our previous experiment. We have repeated the *in planta* assay
with the pathogen inoculation concentration at 10^5 CFU/mL and extended the
observation period to 15 days, by which time the disease index in the control group
reached approximately 82%. Additionally, all qPCR-based pathogen
quantifications have now been converted to CFU/mL to ensure biological
relevance.

Our results demonstrate that the control group exhibited the highest disease
index at 81.8%, and the pathogen density is 1.9×10^5 CFU/mL. The combined
treatment with *F. anhuiense* K5 and tocopherol acetate showed the lowest disease
index (22.7%) and the lowest pathogen density (6.1×10^4 CFU/mL). These findings
indicate that the combination of tocopherol acetate and *F. anhuiense* K5
effectively suppressed *Fusarium* invasion, offering a potential new strategy for

controlling tomato *Fusarium* wilt in hydroponic-based system. Future research will focus on the development of specific control measures, including evaluating the long-term durability of the protection and its effectiveness in complex soil environments, which are beyond the objectives of this study. Nevertheless, our experimental results confirmed that tocopherol acetate significantly enriches *F. anhuiense* K5, and the combined application of tocopherol acetate and *F. anhuiense* K5 effectively controls tomato *Fusarium* wilt. Thank you once again for your insightful comments, which have greatly improved our work.

References

- (1) Li, Y.-T., Hwang, S.-G., Huang, Y.-M., and Huang, C.-H. (2018). Effects of *Trichoderma asperellum* on nutrient uptake and *Fusarium* wilt of tomato. *Crop Protection* 110, 275-282.
- (2) Matthews, A., Muthukumar, S.P.T., Hamill, S., Aitken, E.A.B., and Chen, A. (2023). Impact of inoculum density of *Fusarium oxysporum* f. sp. *zingiberi* on symptomatic appearances and yield of ginger (*Zingiber officinale* Roscoe). *Access Microbiol* 5.
- (3) Jambagi, S., and Dixelius, C. (2023). A robust hydroponic-based system for screening red clover (*Trifolium pratense*) for *Fusarium avenaceum*. *Legume Science* 5, e209.
- (4) Song, W., Zhou, L., Yang, C., Cao, X., Zhang, L., and Liu, X. (2004). Tomato *Fusarium* wilt and its chemical control strategies in a hydroponic system. *Crop Protection* 23, 243-247.

(2) Pathogen density was measured only at 24 and 72 hours, which does not reflect the ecological behavior of *Fusarium*. Typically, *Fusarium* requires more than 7 days to colonize and penetrate plant roots. Therefore, pathogen density should be measured at additional time points over an extended period.

Answer: We have extended the observation period to 15 days, by which point the disease index in the control group had reached 81.8%. We measured the pathogen density at this stage and found that the control group exhibited the highest pathogen load, close to 2×10^5 CFU/mL. In contrast, the combined treatment group showed the lowest pathogen density at 6×10^4 CFU/mL. The treatment of tocopherol acetate and the *F. anhuiense* K5 alone resulted in pathogen densities of 1.3×10^5 CFU/mL and 0.9×10^5 CFU/mL, respectively.

In this study, we did not observe pathogen enrichment effect from tocopherol acetate application. This may be attributed to the enhancement of the plant's defense capacity by tocopherol acetate (Stahl et al., 2019; Ma et al., 2020). Over the extended observation period, the accumulation of antimicrobial secondary metabolites released by the plant likely contribute to the reduction of pathogen density in the rhizosphere (Liu et al., 2024). Furthermore, the higher amount of diseased plant residue in the control group may have released phenolic acids, potentially promoting pathogen proliferation in the control. Concurrently, we confirmed that tocopherol acetate significantly promoted the growth of *F. anhuiense* K5, indicating that the enrichment of *F. anhuiense* K5 by tocopherol acetate is a robust effect, even amidst the complex interactions between

89 **metabolites and microorganisms during the infection process.**

**References**

(1) Ma, J., Qiu, D., Pang, Y., Gao, H., Wang, X., and Qin, Y. (2020). Diverse roles of tocopherol acetates in
response to abiotic and biotic stresses and strategies for genetic biofortification in plants. *Molecular*
*Breeding* 40, 18.

(2) Stahl, E., Hartmann, M., Scholten, N., and Zeier, J. (2019). A Role for tocopherol acetate biosynthesis in
*Arabidopsis* basal immunity to bacterial infection. *Plant Physiology* 181, 1008-1028.

(3) Liu, Y., Esposito, D., Mahdi, L.K., Porzel, A., Stark, P., Hussain, H., Scherr-Henning, A., Isfort, S., Bathe,
U., Acosta, I.F., Zuccaro, A., Balcke, G.U., and Tissier, A. (2024). Hordedane diterpenoid phytoalexins
restrict *Fusarium graminearum* infection but enhance *Bipolaris sorokiniana* colonization of barley roots.
*Molecular Plant* 17, 1307-1327.

(3) More important, the actual disease index and suppression efficacy remain
unaddressed in the manuscript.

**Answer: We calculated the disease index and suppression efficacy for both the**
**control and treatment groups. The results demonstrated that the control group**
**exhibited the highest disease index at 81.8%. In contrast, the treatments with strain**
**K5 or tocopherol acetate alone showed disease index of 39.5% and 38.4%,**
**corresponding to suppression efficacies of 51.7% and 52.9%, respectively. Most**
**notably, the combined treatment of strain K5 and tocopherol acetate achieved the**
**lowest disease index (22.7%) and the highest suppression efficacy (72.3%).**

**The pathogen density corroborated these disease index findings: the control**
**group supported the highest pathogen load, while the combined treatment group**
**showed the lowest. In this repeated experiment, the application of tocopherol**
**acetate alone provided a measurable level of disease control. Although the**
**pathogen density in the tocopherol acetate-treated group was numerically lower**
**than that in the control, the difference was not statistically significant. This**
**observed trend towards reduced disease could be attributed to the tocopherol**
**acetate's potential role in priming the plant's defense system (Stahl et al., 2019;**
**Ma et al., 2020), thereby enhancing its tolerance to the pathogen, even if it did not**
**lead to a significant reduction in pathogen colonization under these experimental**
**conditions.**

**References**

(1) Ma, J., Qiu, D., Pang, Y., Gao, H., Wang, X., and Qin, Y. (2020). Diverse roles of tocopherol acetates in
response to abiotic and biotic stresses and strategies for genetic biofortification in plants. *Molecular*
*Breeding* 40, 18.

(2) Stahl, E., Hartmann, M., Scholten, N., and Zeier, J. (2019). A Role for tocopherol acetate biosynthesis in
*Arabidopsis* basal immunity to bacterial infection. *Plant Physiology* 181, 1008-1028.

**The changes in the manuscript are shown as follows:**

**Result part (line 269 to 284)**

**Effect of *F. anhuiense* and tocopherol acetate on the *in planta* *Fusarium* wilt**

We investigated the effects of *F. anhuiense* K5 and tocopherol acetate, both alone and
in combination, on tomato *in planta* *Fusarium* wilt, with assessments conducted at 15
139 days post-inoculation. The results demonstrated that the control group exhibited the
140 highest disease index, reaching 81.8% (Fig. 6). Treatments with *F. anhuiense* K5 or
141 tocopherol acetate alone significantly reduced the disease index to 39.5% and 38.4%,
respectively, approximately half that of the control. The combined application of *F.*
*anhuiense* K5 and tocopherol acetate resulted in the lowest disease index, recorded at
only 22.7%. The control efficacy of *F. anhuiense* K5, tocopherol acetate, and the
combined treatment was 51.7%, 52.9%, and 72.3%, respectively (Table S18). Pathogen
density quantification in the rhizosphere soil corroborated the disease index findings:
the control group supported the highest pathogen load (1.9×10^5 CFU/mL), whereas the
combined treatment group showed the lowest (6.1×10^4 CFU/mL). Furthermore,
tocopherol acetate significantly increased the abundance of *F. anhuiense* K5, with the
bacterial population in the combined treatment showing a 3.16-fold increase compared
to the treatment with *F. anhuiense* K5 alone.

**Discussion part (line 352 to 363)**

The enrichment of *F. anhuiense* K5 by tocopherol acetate was also demonstrated
by the *in planta* assay (Fig. 6). The combined application of *F. anhuiense* K5 and
tocopherol acetate resulted in the lowest disease index among all treatments. It is
postulated that the underlying mechanism for the enhanced efficacy involves both the

enrichment of *F. anhuiense* K5 by tocopherol acetate and the concomitant tocopherol-
mediated activation of plant immune responses^{35,36}. Collectively, our findings suggest
that the combined application of *F. anhuiense* K5 and tocopherol acetate represents a
promising novel strategy for managing tomato *Fusarium* wilt in hydroponic-based
system. Future research will focus on the development of specific control measures,
including evaluating the long-term durability of the protection and its effectiveness in
complex soil environments, which are beyond the objectives of this study.

**Reviewer #5 (Remarks to the Author):**

The authors have responded positively to my previous comments and have revised the
manuscript accordingly.

Reviewer #5 (Remarks on code availability):

The authors have provided a Zenodo repository containing the code necessary to
reproduce the results.

**Answer: We sincerely thank Reviewer #5 for their positive feedback on our**
**revisions and for acknowledging the availability of our code repository.**

**Reviewer #6 (Remarks to the Author):**

I would like to thank the authors for meticulously addressing all my concerns.

I have no further concerns.

Reviewer #6 (Remarks on code availability):

previously I have tried to run the code and had uttered some concerns. Now all
subdirectories are equipped with respective README files and much improved
installation instructions

**Answer: We are truly grateful to Reviewer #6 for the thoughtful and constructive**
**feedback throughout the review process. We are delighted to hear that the**
**revisions, particularly the improved documentation and installation instructions,**
**have adequately addressed the previous concerns.**
